# Plasma membrane phosphatidylinositol (4,5)-bisphosphate is critical for determination of epithelial characteristics

Kaori Kanemaru[1,10], Makoto Shimozawa [2,10], Manabu Kitamata[2], Rikuto Furuishi[1], Hinako Kayano[2], Yui Sukawa[2], Yuuki Chiba[2], Takatsugu Fukuyama[2], Junya Hasegawa[3,4], Hiroki Nakanishi[5,6], Takuma Kishimoto[7], Kazuya Tsujita [8,9], Kazuma Tanaka[7], Toshiki Itoh[8,9], Junko Sasaki[3,4], Takehiko Sasaki [3,4], Kiyoko Fukami [2✉] & Yoshikazu Nakamura [1✉]

Epithelial cells provide cell-cell adhesion that is essential to maintain the integrity of multi-cellular organisms. Epithelial cell-characterizing proteins, such as epithelial junctional proteins and transcription factors are well defined. However, the role of lipids in epithelial characterization remains poorly understood. Here we show that the phospholipid phosphatidylinositol (4,5)-bisphosphate [PI(4,5)$P_2$] is enriched in the plasma membrane (PM) of epithelial cells. Epithelial cells lose their characteristics upon depletion of PM PI(4,5)$P_2$, and synthesis of PI(4,5)$P_2$ in the PM results in the development of epithelial-like morphology in osteosarcoma cells. PM localization of PARD3 is impaired by depletion of PM PI(4,5)$P_2$ in epithelial cells, whereas expression of the PM-targeting exocyst-docking region of PARD3 induces osteosarcoma cells to show epithelial-like morphological changes, suggesting that PI(4,5)$P_2$ regulates epithelial characteristics by recruiting PARD3 to the PM. These results indicate that a high level of PM PI(4,5)$P_2$ plays a crucial role in the maintenance of epithelial characteristics.

[1] Department of Applied Biological Science, Faculty of Science and Technology, Tokyo University of Science, Noda, Chiba, Japan. [2] Laboratory of Genome and Biosignals, School of Life Sciences, Tokyo University of Pharmacy and Life Sciences, Hachioji, Tokyo, Japan. [3] Department of Biochemical Pathophysiology, Medical Research Institute, Tokyo Medical and Dental University, Bunkyo-ku, Tokyo, Japan. [4] Department of Lipid Biology, Graduate School of Medical and Dental Sciences, Tokyo Medical and Dental University, Bunkyo-ku, Tokyo, Japan. [5] Research Center for Biosignal, Akita University, Akita-city, Akita, Japan. [6] Lipidome Lab Co., Ltd, Akita-city, Akita, Japan. [7] Division of Molecular Interaction, Institute for Genetic Medicine, Hokkaido University Graduate School of Life Science, Sapporo, Hokkaido, Japan. [8] Biosignal Research Center, Kobe University, Kobe, Hyogo, Japan. [9] Division of Membrane Biology, Department of Biochemistry and Molecular Biology, Kobe University Graduate School of Medicine, Kobe, Hyogo, Japan. [10]These authors contributed equally: Kaori Kanemaru, Makoto Shimozawa. ✉email: kfukami@toyaku.ac.jp; ynakamur@rs.tus.ac.jp

Epithelial cells provide cell-cell adhesion that is essential for maintaining the integrity of multicellular organisms and to ensure that epithelial tissues function as a covering or lining for body surfaces, such as the skin, gut, and ducts. Many of the physical properties of epithelial cells rely on their attachment to each other, which is mediated by several types of cell junctions, including tight junctions and adherens junctions. In mammals, epithelialization of the developing embryo occurs early during the compaction of the blastula[1, 2]. Nonetheless, even after epithelialization, epithelial cells still possess the plasticity to differentiate into mesenchymal cells. Shortly after the epithelialization of the blastula, epithelial cells lose their characteristics to form the primary mesenchyme during gastrulation[3, 4]. This process, whereby epithelial characteristics are lost and mesenchymal features are gained, is known as epithelial-mesenchymal transition (EMT). The EMT, which is observed during embryonic development, is aberrantly activated under pathological conditions, including organ fibrosis and cancer[5]. The loss of epithelial characteristics facilitates the dissociation of tumor cells from the primary tumor and their dissemination into blood circulation, both of which are critical events for tumor metastasis[6]. Therefore, it is important to understand epithelial characteristics and identify the factors that determine or regulate them to develop strategies to suppress tumor development and metastasis. Epithelial cell-characterizing proteins, such as epithelial junctional proteins and transcription factors, are well defined[7]. However, the role of lipids in epithelial characterization remains poorly understood.

Phosphatidylinositol (PI), is a phospholipid that can undergo phosphorylation, generating seven phosphorylated forms known as phosphatidylinositol phosphates (PIPs). Among PIPs, phosphatidylinositol 4,5-bisphosphate [PI(4,5)P$_2$] is generated by the phosphorylation of the inositol moiety of PI at positions 4 and 5. PI(4,5)P$_2$ is synthesized mainly by phosphorylation of PI by PI 4-kinases, forming phosphatidylinositol 4-phosphate [PI(4)P], which is then phosphorylated by PI(4)P 5-kinase (PIP5K) in the plasma membrane (PM) to generate PI(4,5)P$_2$. PI(4,5)P$_2$ serves as a substrate for phospholipase C (PLC) and as a source for the second messengers, inositol 1,4,5-trisphosphate (IP$_3$) and diacylglycerol (DG)[8, 9]. In addition, PI(4,5)P$_2$ is phosphorylated by class I phosphoinositide-3 kinase (PI3K) to generate PI(3,4,5)P$_3$, an important signaling lipid that regulates cell proliferation, survival, and migration[10]. In addition to its role as a second messenger-generating lipid, PI(4,5)P$_2$ also regulates the functions of various proteins, including actin-, endocytosis-, and exocytosis-regulating proteins[11–13].

In this study, we demonstrate that PI(4,5)P$_2$ is enriched in the PM of epithelial cells and plays a critical role in the maintenance and acquisition of epithelial characteristics.

## Results

**PI(4,5)P$_2$ is enriched in the PM of epithelial cells**. Our previous studies revealed that PI(4,5)P$_2$-metabolizing enzymes play a critical role in the maintenance of normal skin functions[14, 15]. Hence, we explored the amount and localization of PI(4,5)P$_2$ in mouse skin using immunofluorescence with an anti-PI(4,5)P$_2$ monoclonal antibody[16]. The intensity of the PI(4,5)P$_2$ signal was much stronger in the epidermis than in the dermis (Fig. 1a). While the epidermis is mainly composed of epithelial cells (keratinocytes), the dermis is mainly composed of non-epithelial cells. Therefore, we hypothesized that the amount of PI(4,5)P$_2$ may differ between epithelial and non-epithelial cells. To assess this hypothesis, PI(4,5)P$_2$ was detected using immunofluorescence in epithelial and non-epithelial cell lines. The intensity of the PI(4,5)P$_2$ signal was much stronger in the PM of epithelial cells such as HaCaT and NMuMG cells, compared to that of non-epithelial

cells such as human dermal fibroblasts (HDFs), Swiss3T3, and U2OS cells (Fig. 1b, c and Supplementary Fig. 1a). In addition, the pleckstrin homology (PH) domain of PLCδ1, which specifically interacts with PI(4,5)P$_2$, was more enriched in the PM of HaCaT cells than in HDF cells (Supplementary Fig. 1b, c). Furthermore, the intensity of the PI(4,5)P$_2$ signal was attenuated when HaCaT cells lost their epithelial characteristics, such as PM localization of junctional and polarity proteins, upon TGFβ1 treatment (Fig. 1d, e, Supplementary Fig. 1d, e) or upon overexpression of SNAI1 or SNAI2 (Fig. 1f, g, Supplementary Fig. 1f, g). Reverse-phase LC-MS/MS revealed that the level of phosphatidylinositol bisphosphate (PIP$_2$) was higher in untreated HaCaT cells than in TGFβ1-treated HaCaT and HDF cells (Fig. 1h). Since PI(4,5)P$_2$ is much more abundant than the other two regioisomers of PIP$_2$, namely, PI(3,4)P$_2$ and PI(3,5)P$_2$, in mammalian cells[17], the lower PIP$_2$ levels observed in TGFβ1-treated HaCaT and HDF cells were most likely attributable to the reduction of PI(4,5)P$_2$ levels. Among the PIP$_2$ molecular species, the amount of PIP$_2$ with zero, one, or two double bonds in acyl chains was higher in untreated HaCaT cells than in TGFβ1-treated HaCaT cells and HDF cells, suggesting that PI(4,5)P$_2$ with high lipid saturation status was enriched in untreated HaCaT cells compared to non-epithelial cells (Fig. 1i). In addition, the amounts of phosphatidylinositol (PI) and phosphatidylinositol monophosphate (PIP) were also higher in epithelial cells than in TGFβ1-treated HaCaT and HDF cells (Supplementary Fig. 1h). In contrast, the levels of phosphatidylcholine (PC) were not significantly different between TGFβ1-treated HaCaT and HDF cells and untreated HaCaT cells (Supplementary Fig. 1h). Taken together, these results indicate an enrichment of PI(4,5)P$_2$ in the PM of epithelial cells compared to that of non-epithelial cells.

**PI(4,5)P$_2$ depletion abolishes to HaCaT epitheliality**. Since the PI(4,5)P$_2$ level was higher in epithelial cells, we hypothesized that PI(4,5)P$_2$ might play a role in the maintenance of epithelial characteristics. To test this possibility, PI(4,5)P$_2$ was depleted in HaCaT cells. To achieve this, we generated a PM-targeting PI(4,5)P$_2$ phosphatase in which the 5-phosphatase domain of INPP5E was fused with GFP and with the PM-targeting myristoylation sequence of Lyn (hereinafter designated as Lyn-INPP5Ewt-GFP), followed by being expressed in HaCaT cells. HaCaT cells expressing GFP or Lyn-INPP5Emt-GFP with an inactivated 5-phosphatase domain of INPP5E were used as negative controls. As expected, expression of Lyn-INPP5Ewt-GFP attenuated the PM PI(4,5)P$_2$ signal intensity in HaCaT cells and affected the morphology of epithelial cells (Fig. 2a, b, Supplementary Fig. 2a). GFP- or Lyn-INPP5Emt-GFP-expressing HaCaT cells had a typical polygonal cell shape, whereas Lyn-INPP5Ewt-GFP-expressing HaCaT cells did not present typical epithelial morphology and displayed cell spreading (Fig. 2a, bottom panels), in addition to increased cell area (Fig. 2c); such alterations were also observed upon treatment with TGFβ1, which also resulted in cell spreading and increased cell area (Fig. 2c). Changes in cell morphology by PI(4,5)P$_2$ depletion corresponded with the reorganization of the actin cytoskeleton. Cortical organization of actin filaments is a hallmark of epithelial cells. In PI(4,5)P$_2$-depleted HaCaT cells, the intensity of cortical actin signal was attenuated and F-actin was assembled into actin stress fibers, which is a characteristic of mesenchymal cells (Fig. 2a). In addition, PM localization of junctional and polarity proteins was impaired by PI(4,5)P$_2$ depletion (Fig. 2d, e). Nonetheless, PI(4,5)P$_2$ depletion did not alter the expression levels of junctional and polarity proteins (Fig. 2f), suggesting that it induced the mislocalization of these proteins. We further examined the effect of PI(4,5)P$_2$ depletion

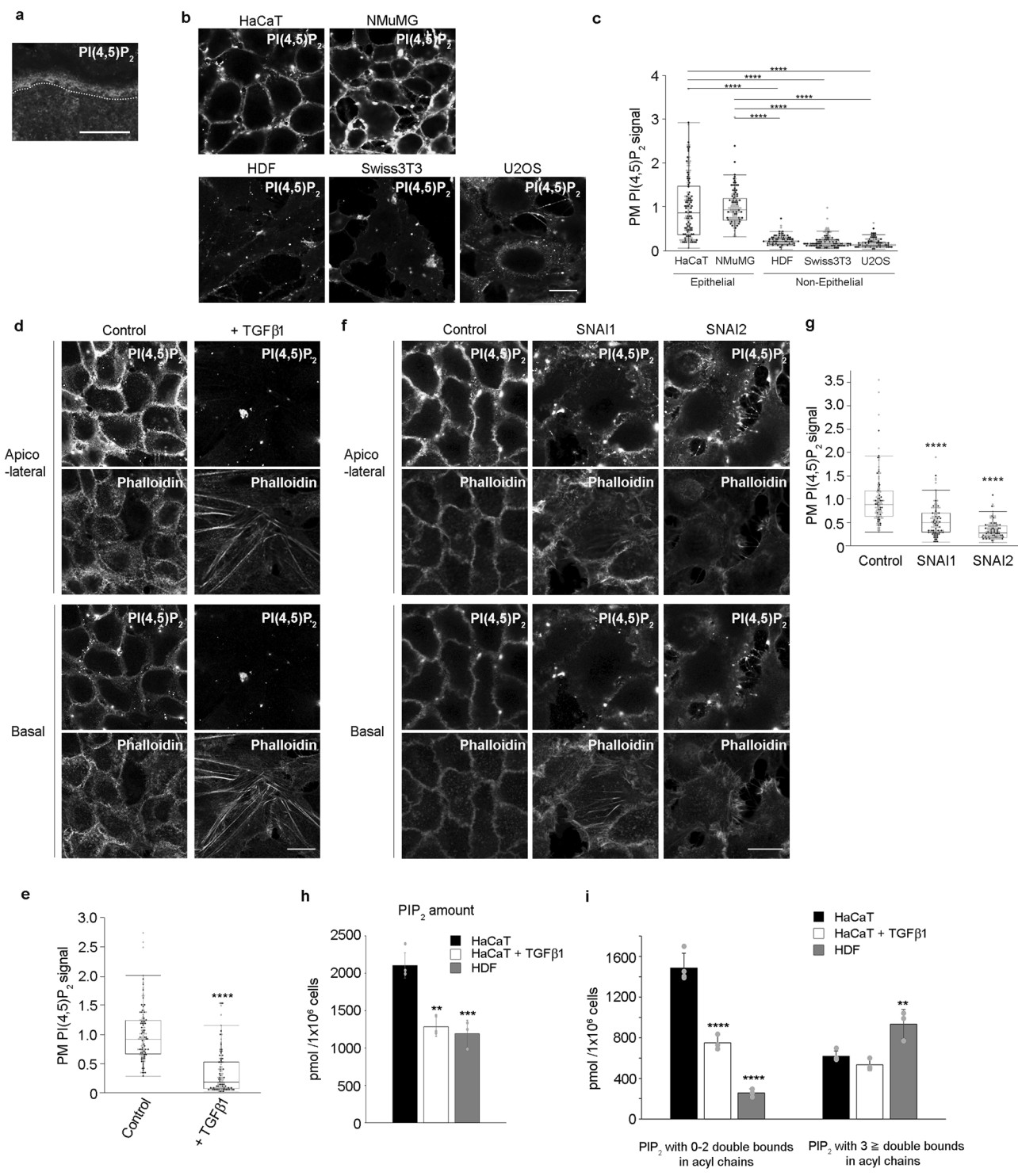

on gene expression in HaCaT cells. Although $PI(4,5)P_2$ depletion did not decrease the expression level of the epithelial cell-enriched gene *CDH1*, it induced the expression of mesenchymal cell-enriched genes, including *CDH2*, which have been reported as EMT core genes[18] (Fig. 2g and Supplementary Fig. 2b). However, $PI(4,5)P_2$ depletion was not accompanied by the expression of *VIM*, a marker characteristic of mesenchymal cells, indicating that $PI(4,5)P_2$ depletion did not induce complete EMT (Supplementary Fig. 2b). Moreover, the expression levels of *SNAI1* and *SNAI2* were unaffected by $PI(4,5)P_2$ depletion (Supplementary Fig. 2c), indicating that $PI(4,5)P_2$ depletion did not induce the loss of epithelial characteristics

through the induction of EMT-inducing transcription factors. Since Lyn-INPP5Ewt-GFP dephosphorylates PM $PI(4,5)P_2$ to generate PI(4)P, expression of Lyn-INPP5Ewt-GFP may lead to the accumulation of PI(4)P, which induces the loss of epithelial characteristics. However, phosphoinositide regioisomer measurement by chiral column chromatography and mass spectrometry[19] revealed that PI(4)P did not accumulate and that its levels decreased in Lyn-INPP5Ewt-GFP-expressing cells (Supplementary Fig. 2d). Taken together, these results suggest that in the PM, $PI(4,5)P_2$ plays a critical role in the maintenance of epithelial characteristics, as well as that $PI(4,5)P_2$ depletion in epithelial cells leads to the loss of epithelial characteristics.

**Fig. 1 PI(4,5)P$_2$ is enriched in the plasma membrane of epithelial cells. a** Immunofluorescence detection of PI(4,5)P$_2$ in skin sections of newborn mice. The dotted line denotes the border between the epidermis and the dermis. Results shown are representative of two independent experiments. **b, c** Immunofluorescence detection of PI(4,5)P$_2$ in epithelial (HaCaT and NMuMG) and non-epithelial (HDF, Swiss3T3, and U2OS) cell lines **b**. Quantification of apicolateral plasma membrane (PM) PI(4,5)P$_2$ signals **c**. In total, 138 HaCaT, 122 NMuMG, 104 HDF, 120 Swiss3T3, 102 U2OS cells were examined over two independent experiments. **d–g** Immunofluorescence detection of PI(4,5)P$_2$ in HaCaT cells treated with or without TGFβ1 for 72 h **d** and HaCaT cells overexpressing SNAI1 or SNAI2 **f**. F-actin was visualized using phalloidin. Quantification of apicolateral PM PI(4,5)P$_2$ signals **e, g**. In total, 120 control and 109 TGFβ1-treated cells were examined over two independent experiments **e**. In total, 100 control, 102 SNAI1-, and 115 SNAI2-expressing cells were examined over two independent experiments **g**. **h, i** PIP$_2$ amount in HaCaT cells, treated with or without TGFβ1 for 72 h, and HDF cells was determined using mass spectrometry **h**. The amount of PIP$_2$ with zero, one, or two double bonds and three or more double bonds in acyl chains is also shown **i**. N = 4 for untreated HaCaT cells, and N = 3 for TGFβ1-treated HaCaT and HDF cells. Data are represented as mean ± SD **h, i**. ***p = 0.0006, **p = 0.0012 **h**, 0.0061 **i** (versus HaCaT cells) **h, i**. The box plots are presented with the elements: center line, median; box limits, Q1 and Q3; whiskers, 1.5× interquartile range. Outliers are also shown. Individual data points are displayed [gray points, data from the first experiments; black points, data from the second experiments **c, e, g**]. Significance was tested using one-way ANOVA with Tukey-Kramer's post hoc test **c, g, h, i** and the two-sided Welch's t-test **e**. ****p < 0.0001 [versus HaCaT or NMuMG cells **c**, control cells **e, g**, and HaCaT cells **h, i**]. Scale bar = 30 μm **a** and 20 μm **b, d, f**. Source data are provided as a Source Data file.

## PIP5K1A maintains HaCaT epitheliality.

Given that the PM PI(4,5)P$_2$ levels were higher in epithelial cells than in non-epithelial cells and that PI(4,5)P$_2$ plays a role in the maintenance of epithelial characteristics, PI(4,5)P$_2$-producing enzymes might contribute to the maintenance of epithelial characteristics by ensuring high PI(4,5)P$_2$ levels. Since PI(4,5)P$_2$ is mainly produced by phosphorylation of PI(4)P by PIP5K, we examined the mRNA expression level of PIP5Ks in HaCaT cells with or without TGFβ1 treatment. TGFβ1 treatment reduced the expression levels of human *PIP5K1A* (*hPIP5K1A*) and *hPIP5K1B* but not that of *hPIP5K1C* (Fig. 3a). Additionally, we examined the localization of hPIP5K1A fused with GFP (hPIP5K1A-GFP). hPIP5K1A-GFP was more abundant in the PM of HaCaT cells than in the PM of non-epithelial HDF cells (Supplementary Fig. 3a, c). In addition, localization of hPIP5K1A-GFP to the PM was attenuated in HaCaT cells after TGFβ1 treatment (Supplementary Fig. 3b, c). Since *hPIP5K1A* mRNA was mostly decreased by TGFβ1 treatment and localization of hPIP5K1A-GFP was different between epithelial and non-epithelial cells, hPIP5K1A might contribute to the maintenance of high PI(4,5)P$_2$ levels in the epithelial cells. To test this, *hPIP5K1A* was silenced using two siRNAs targeting different parts of the *hPIP5K1A* transcript (Fig. 3b). hPIP5K1A depletion remarkably decreased the intensity of the PI(4,5)P$_2$ signal in the PM of HaCaT cells (Fig. 3c, d). Consistent with a decreased level of PI(4,5)P$_2$, hPIP5K1A-depleted HaCaT cells lost typical epithelial characteristics and displayed cell spreading with actin stress fibers and increased cell area (Fig. 3c–g). Furthermore, *CDH2* was upregulated upon siRNA-mediated depletion of hPIP5K1A (Fig. 3h). siRNA-resistant hPIP5K1A-GFP-expressing cells did not lose epithelial characteristics upon introduction of siRNA targeting *hPIP5K1A* (Supplementary Fig. 3d-f). These results strongly suggest that the PI(4,5)P$_2$-producing enzyme hPIP5K1A contributes to the maintenance of PI(4,5)P$_2$ levels and the epithelial characteristics in epithelial cells.

## Effects of PI3K and PLC inhibition on HaCaT epitheliality.

Since PI(4,5)P$_2$ is phosphorylated by PI3K to produce PI(3,4,5)P$_3$, PI(4,5)P$_2$ depletion may affect epithelial characteristics by reducing the amount of PI(3,4,5)P$_3$ and impairing the signaling pathways downstream of PI(3,4,5)P$_3$. To address this, we examined the effect of PI(3,4,5)P$_3$ depletion on the maintenance of epithelial characteristics. PI(3,4,5)P$_3$ depletion by inhibition of PI(3,4,5)P$_3$-producing enzyme PI3K with LY294002 treatment did not affect the actin cytoskeleton and only slightly increased the cell area in HaCaT cells (Supplementary Fig. 4a). We also confirmed that inhibition of Akt, one of the main downstream effectors of PI(3,4,5)P$_3$, did not induce loss of epithelial characteristics (Supplementary Fig. 4b). These results strongly suggest

that the impairment of PI(3,4,5)P$_3$ downstream signaling did not contribute to the loss of epithelial characteristics upon PI(4,5)P$_2$ depletion. PI(4,5)P$_2$ is hydrolyzed by PLC to generate the second messengers IP$_3$ and DG. Therefore, depletion of PI(4,5)P$_2$ might decrease the production of these second messengers, leading to the loss of epithelial characteristics. To test this possibility, we inhibited the production of these second messengers by treating HaCaT cells with the PLC inhibitor U73122 or by silencing *PLCD1* (*PLCδ1*), which is the predominant PLC isozyme in keratinocytes[14]. PLC inhibition upon U73122 treatment or knockdown of PLCδ1 did not induce changes in the actin cytoskeleton or increase the cell area of HaCaT cells (Supplementary Fig. 4c, d), suggesting that a decrease in second messengers did not lead to the loss of epithelial characteristics. These results strongly suggest that PI(4,5)P$_2$-derived signaling molecules do not contribute to the loss of epithelial characteristics in PI(4,5)P$_2$-depleted HaCaT cells.

## PI(4,5)P$_2$ partly suppresses the loss of epitheliality.

The levels of PI(4,5)P$_2$ were higher in epithelial cells than in non-epithelial cells (Fig. 1). In addition, PI(4,5)P$_2$ depletion led to the loss of epithelial characteristics in epithelial cells (Figs. 2 and 3). These results strongly suggest that PI(4,5)P$_2$ plays a critical role in the maintenance of epithelial characteristics. Therefore, we examined whether TGFβ1-induced loss of epithelial characteristics was suppressed by increasing PI(4,5)P$_2$ levels. To generate PI(4,5)P$_2$ in the PM, the PI(4,5)P$_2$-producing enzyme mouse PIP5K1C635 (mPIP5K1C) was fused with the PM-targeting myristoylation sequence of Lyn and GFP, generating a construct hereinafter designated as Lyn-mPIP5Kwt-GFP. Lyn-mPIP5Kwt-GFP was overexpressed in HaCaT cells. Lyn-mPIP5Kmt-GFP with kinase-dead mPIP5K1C was used as a negative control. The results of immunofluorescence analysis revealed that after TGFβ1 treatment, the intensity of the PI(4,5)P$_2$ signal was very weak in GFP- or Lyn-mPIP5Kmt-GFP-expressing HaCaT cells, whereas it was still detected in the PM of Lyn-mPIP5Kwt-GFP-expressing HaCaT cells (Fig. 4a, b). After TGFβ1 treatment, GFP- or Lyn-mPIP5Kmt-GFP-expressing cells lost typical epithelial characteristics and exhibited mesenchymal-like morphology with increased cell area and actin stress fibers, whereas Lyn-mPIP5Kwt-GFP-expressing cells still showed epithelial characteristics (Fig. 4a, c, d). In addition, upregulation of *CDH2* by TGFβ1 treatment was partly suppressed in Lyn-mPIP5Kwt-GFP-expressing cells (Fig. 4e). Thus, elevation of PI(4,5)P$_2$ levels partly suppressed the loss of epithelial characteristics caused by TGFβ1 treatment.

## Depletion of PI(4,5)P$_2$ decreases the PM cholesterol level.

Since a decrease in the PM cholesterol level disturbs the localization of

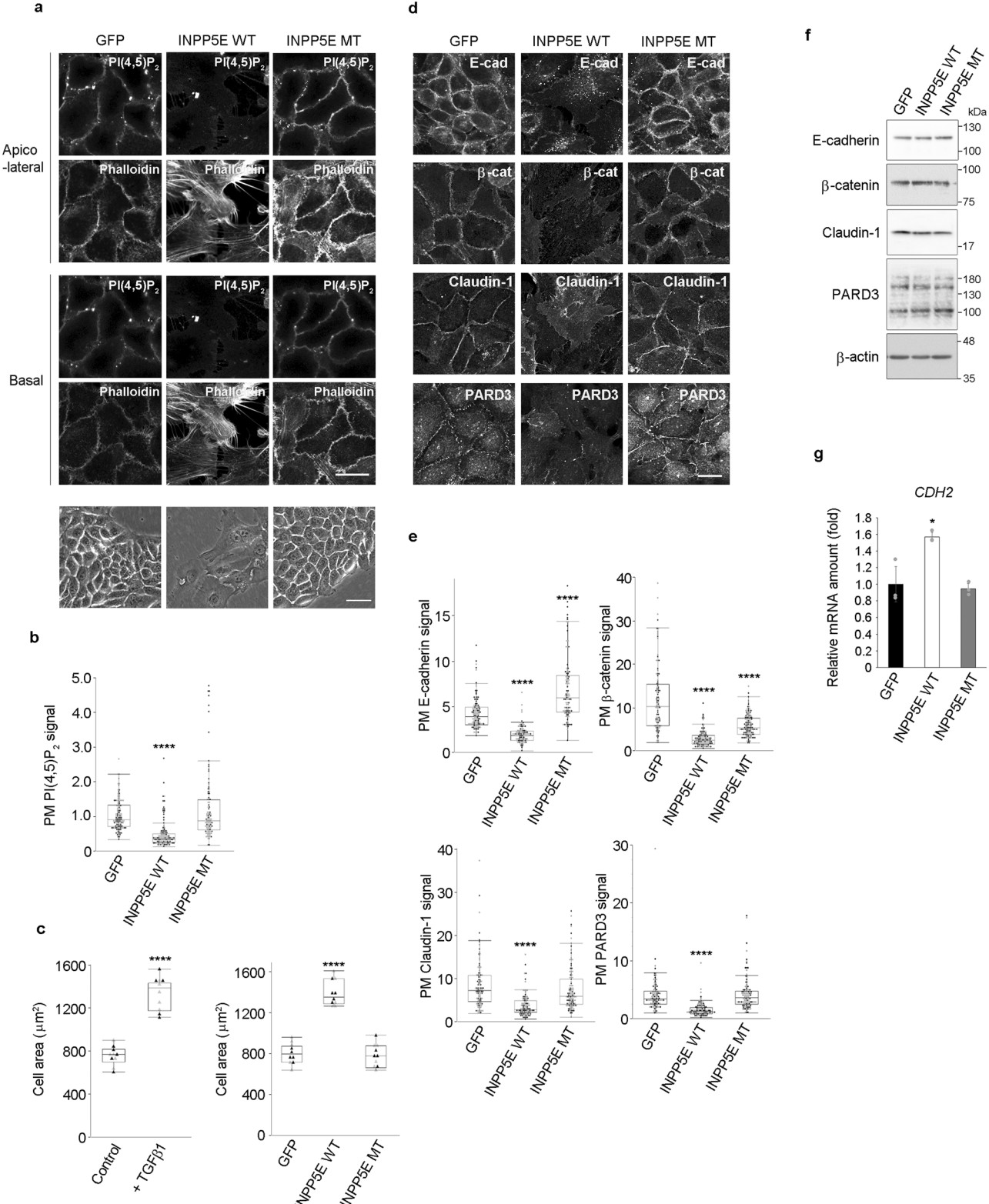

epithelial junctional proteins[20], as does PI(4,5)P$_2$ depletion, it is possible that the abundance of cholesterol in the PM might be affected by PI(4,5)P$_2$ depletion. Cholesterol visualization using a mCherry-D4 (a recombinant protein of perfringolysin O-derived cholesterol-binding probe), or filipin, revealed that the abundance of cholesterol in the PM was drastically decreased upon PI(4,5)P$_2$ depletion by expression of Lyn-INPP5Ewt-GFP (Fig. 5a, b, Supplementary Fig. 5a, b). Additionally, the concentrations of

cholesterol and cholesterol ester were quantified using an enzyme-linked immunosorbent assay. The results of the analysis revealed that the amount of cholesterol and cholesterol ester did not decrease after PI(4,5)P$_2$ depletion (Fig. 5c), suggesting that PI(4,5)P$_2$ did not decrease the total amount of cholesterol and cholesterol ester but affected their transport to the PM or their extraction. Since the levels of PI(4,5)P$_2$ were different between epithelial and non-epithelial cells, we also compared PM

**Fig. 2 HaCaT cells lose their epithelial characteristics upon PI(4,5)P$_2$ depletion. a, b** Detection of PI(4,5)P$_2$ and F-actin (phalloidin) in GFP-, Lyn-INPP5Ewt-GFP (INPP5E WT)-, and Lyn-INPP5Emt-GFP (INPP5E MT)-expressing HaCaT cells. To observe actin stress fibers, F-actin images (phalloidin) at the basal plane of the top panels are shown. Phase images are also shown **a**. Quantification of the apicolateral plasma membrane (PM) PI(4,5)P$_2$ signal **b**. In total, 120 GFP-, 117 INPP5E WT-, and 100 INPP5E MT-expressing cells were examined over two independent experiments **b**. **c** Cell areas of HaCaT cells treated with or without TGFβ1 for 72 h. Cell areas of GFP-, INPP5E WT-, and INPP5E MT-expressing HaCaT cells are shown. The average cell area was calculated by analyzing 10 distinct fields of view over two independent experiments. **d, e** Immunofluorescence detection of E-cadherin (E-cad), β-catenin (β-cat), claudin-1, and PARD3 in GFP-, INPP5E WT-, and INPP5E MT-expressing HaCaT cells **d**. Images were taken from different fields of view. Quantification of the PM signals of E-cadherin, β-catenin, claudin-1, and PARD3 **e**. In total, 111 GFP-, 96 INPP5E WT-, and 103 INPP5E MT-expressing cells were examined over two independent experiments for E-cadherin. In total, 100 GFP-, 114 INPP5E WT-, and 124 INPP5E MT-expressing cells were examined over two independent experiments for β-catenin. In total, 102 GFP-, 96 INPP5E WT-, and 109 INPP5E MT-expressing cells were examined over two independent experiments for claudin-1. In total, 100 GFP-, 96 INPP5E WT-, and 109 INPP5E MT-expressing cells were examined over two independent experiments for PARD3 **e**. **f** Immunoblotting for E-cadherin, β-catenin, claudin-1, and PARD3 in GFP-, INPP5E WT-, or INPP5E MT-expressing HaCaT cells. 180 kDa, 150 kDa, and 100 KDa forms of PARD3 were detected. β-actin was used as the loading control. Immunoblot data shown are representative of two independent experiments with similar results. **g** Real-time RT-PCR analysis of *CDH2* expression levels in GFP-, INPP5E WT-, and INPP5E MT-expressing HaCaT cells. $N = 3$ for each group. Data are represented as mean ± SD **g**. The box plots are presented with the elements: center line, median; box limits, Q1 and Q3; whiskers, 1.5× interquartile range. Outliers are also shown. Individual data points are displayed [gray points, data from the first experiments; black points, data from the second experiments **b, c, e**]. Significance was tested using one-way ANOVA with Tukey-Kramer's post hoc test **b, e, g** and the two-sided Welch's *t*-test **c**. ****$p < 0.0001$, *$p = 0.0118$ (versus GFP-expressing cells or control cells). Scale bars = 20 μm (except for the bottom panels of **a**) or 50 μm (bottom panels of **a**). Source data are provided as a Source Data file.

cholesterol levels between these cell types. As detected in case of the levels of PI(4,5)P$_2$, the intensity of mCherry-D4 or filipin signal of PM cholesterol was stronger in epithelial cells than in non-epithelial cells (Fig. 5d, e, Supplementary Fig. 5c, d). When HaCaT cells were treated with TGFβ1, the intensity of the cholesterol signal from mCherry-D4 or filipin decreased, similar to the PI(4,5)P$_2$ signal (Fig. 5f, g, Supplementary Fig. 5e, f). The decrease in the PI(4,5)P$_2$ intensity preceded the reduction of cholesterol in the PM (Supplementary Fig. 5e, f), suggesting that PI(4,5)P$_2$ plays a role in the maintenance of cholesterol levels in the PM of epithelial cells. As the decrease in PM cholesterol may contribute to the loss of epithelial characteristics, we examined the effects of depletion of PM cholesterol in HaCaT cells using U-18666A, a compound that blocks cholesterol efflux from the late endosome/lysosome to other organelles, including the PM. Cholesterol depletion from the PM in HaCaT cells induced morphological changes in the cells, which transitioned from a compact polygonal shape to an elongated shape (Fig. 5h). Additionally, PM localization of junctional proteins and a polarity protein were disturbed upon PM cholesterol deletion (Fig. 5h, i). Oxysterol-binding protein-related protein 2 (OSBPL2) is involved in the regulation of cholesterol transport to PM[21]. Knockdown of OSBPL2 decreased PM cholesterol levels and induced loss of epithelial characteristics (Supplementary Fig. 5g–i) in a similar manner as U-18666A treatment. Collectively, these results indicate that depletion of PM cholesterol leads to the loss of epithelial characteristics and strongly suggest that decreased levels of PM cholesterol contribute to the loss of epithelial characteristics upon PI(4,5)P$_2$ depletion.

**PI(4,5)P$_2$ is proximal to epithelial junctional proteins**. To elucidate the mechanisms by which PI(4,5)P$_2$ regulates epithelial characteristics, we screened proteins in close proximity to PI(4,5)P$_2$ in epithelial cells. We established HaCaT cells stably expressing V5 and APEX2 fused to the PH domain of PLCδ1, which specifically binds to PI(4,5)P$_2$ and is localized to the PM (Supplementary Fig. 6a). In the presence of hydrogen peroxide and biotin-phenol, proteins proximal to APEX2 (generally within 30 nm) are biotinylated by APEX2, allowing for their enrichment using streptavidin beads[22]. Detection of biotinylated proteins with streptavidin revealed that biotinylated proteins were mainly located in the PM (Supplementary Fig. 6b). In combination with stable isotope labeling by amino acids in cell culture (SILAC), 99 proteins were identified as proteins proximal to PI(4,5)P$_2$

(Supplementary Data 1). APEX2-fused non-PI(4,5)P$_2$-binding mutant of the PH domain of PLCδ1 (R40L) was used as the negative control. Furthermore, we excluded the proteins that were reported to be proximal proteins of an irrelevant transmembrane protein in a similar proximal biotinylation-based approach with keratinocytes[23]. Known PI(4,5)P$_2$-binding proteins, such as MARCKS, ezrin, cortactin, and IQGAP1, were identified as PI(4,5)P$_2$ proximal proteins, thus confirming that the followed approach is effective for the identification of proteins proximal to PI(4,5)P$_2$. Gene ontology term analysis revealed that proteins related to extracellular exosome, nucleosome, nuclear nucleosome, cell-cell adherens junction, focal adhesion, and cell–cell junction were enriched in the PI(4,5)P$_2$ proximal proteins (Supplementary Fig. 6c). Consistent with the fact that proteins related to cell-cell adherens junction and cell–cell junction are enriched in the PI(4,5)P$_2$ proximal proteins, PI(4,5)P$_2$ was required for the normal localization of junctional proteins (Figs. 2d, e, 3f, g), suggesting that PI(4,5)P$_2$ maintains epithelial characteristics by regulating the localization of junctional proteins.

**PARD3 silencing phenocopies depletion of PM PI(4,5)P$_2$**. PM localization of PARD3 was impaired by PI(4,5)P$_2$ depletion (Figs. 2d, e, 3f, g). Since PARD3 has been shown to associate with PI(4,5)P$_2$[24], PARD3 might be the PI(4,5)P$_2$ downstream effector for the regulation of epithelial characteristics. To test this possibility, *PARD3* expression was silenced using two siRNAs targeting different parts of the *PARD3* transcript. Since PARD3 knockdown induces apoptosis in epithelial cells[25], cells were cultured in the presence of the caspase inhibitor Z-VAD-FMK to prevent apoptosis. PARD3 depletion impaired PM localization of junctional proteins in HaCaT cells (Fig. 6a–c). In addition, PARD3 depletion decreased PM cholesterol in a similar manner to PI(4,5)P$_2$ depletion (Fig. 6b, c). These data indicate that PARD3 silencing phenocopied depletion of PM PI(4,5)P$_2$ and strongly suggest that PI(4,5)P$_2$ regulates epithelial characteristics by ensuring the presence of PARD3 at the PM.

**PI(4,5)P$_2$ partly confers epitheliality on osteosarcoma cells**. Next, we examined the effects of elevation of PI(4,5)P$_2$ levels on non-epithelial cells that have a lower level of PI(4,5)P$_2$. When the PI(4,5)P$_2$ level was elevated in human osteosarcoma U2OS cells by expressing Lyn-mPIP5Kwt-GFP, the cells showed epithelial-like compact morphology and developed cell-cell adhesion

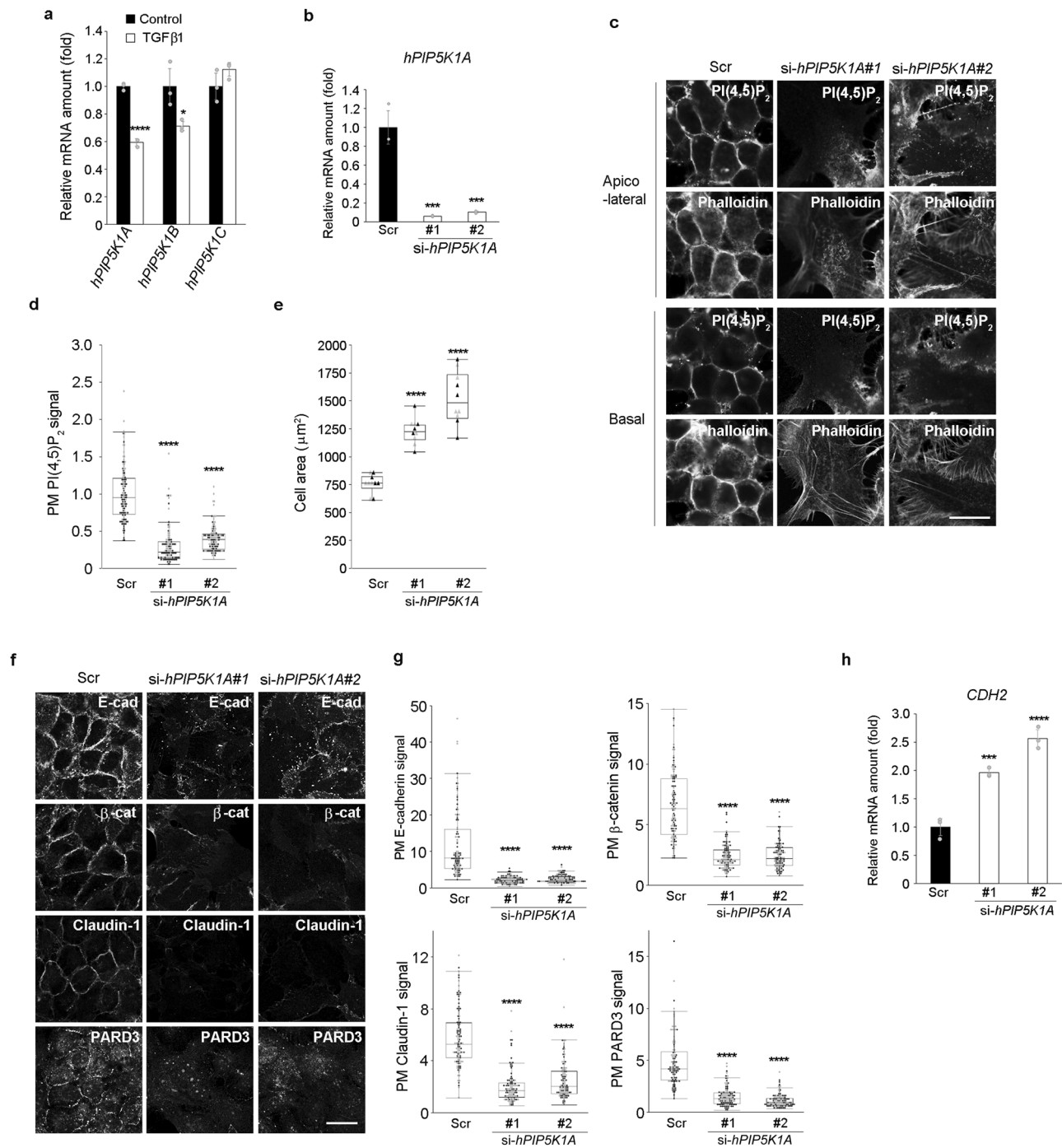

structures (Fig. 7a–c). Elevation of the PI(4,5)P$_2$ level was found to induce epithelial-like morphological changes in MG-63 cells, another osteosarcoma cell line (Supplementary Fig. 7a, b). In addition, in U2OS cells there was an increase in PM cholesterol upon elevation of PI(4,5)P$_2$ levels (Fig. 7a, d). Moreover, CDH2 (N-cadherin), but not CDH1 (E-cadherin), was found to accumulate in the cell-cell contacts in U2OS cells (Fig. 7e, f, i), suggesting that N-cadherin plays a role in the formation of cell-cell junctions in Lyn-mPIP5Kwt-GFP-expressing U2OS cells. Importantly, elevation of PI(4,5)P$_2$ levels induced the accumulation of PARD3 in the PM (Fig. 7g–i). Since PARD3 functions as an exocyst receptor and regulates cell surface transport of junctional proteins[25], we examined whether PM localization of the exocyst-docking region of PARD3 was sufficient to induce the

partial development of epithelial characteristics in U2OS cells. We introduced the exocyst-docking region of human PARD3 [hPARD3 (710–1089)] fused with the PM-targeting myristoylation sequence of Lyn and V5 [Lyn-hPARD3 (710–1089)-V5] into U2OS cells. Lyn-hPARD3 (710–1089)-V5-expressing U2OS cells showed epithelial-like morphology and N-cadherin accumulation in the cell-cell contacts, thereby developing cell-cell adhesion structures in a similar manner to Lyn-mPIP5Kwt-GFP-expressing U2OS cells (Fig. 7j–l). These results strongly suggest that elevation of PI(4,5)P$_2$ level causes osteosarcoma cells acquire epithelial characteristics, in terms of cell morphology, by inducing the localization of PARD3 to the PM.

Next, we examined whether the development of epithelial morphology induced by elevation of PI(4,5)P$_2$ level affected cell

**Fig. 3 PIP5K1A contributes to the maintenance of epithelial characteristics in HaCaT cells. a, b** Real-time RT-PCR analysis of *hPIP5K1A*, *hPIP5K1B*, and *hPIP5K1C* expression in HaCaT cells treated with or without TGFβ1 for 72 h **a** and *hPIP5K1A* mRNA expression in HaCaT cells treated with scrambled (Scr) or *hPIP5K1A*-targeting (#1 and #2) siRNAs **b**. $N = 3$ for each group. ***$p = 0.0002$ (PIP5K1A#1), ***$p = 0.0003$ (PIP5K1A#2) (versus HaCaT cells treated with scrambled siRNA). **c, d** Detection of PI(4,5)P$_2$ and F-actin (phalloidin) in HaCaT cells treated with scrambled (Scr) or *hPIP5K1A*-targeting (PIP5K1A#1 and #2) siRNAs **c**. Quantitation of the apicolateral plasma membrane (PM) PI(4,5)P$_2$ signals **d**. In total, 116 Scr, 112 PIP5K1A#1, and 115 PIP5K1A#2 cells were examined over two independent experiments **d**. **e** Cell areas of HaCaT cells treated with scrambled (Scr) or *hPIP5K1A*-targeting (#1 and #2) siRNAs. The average cell area was calculated by analyzing 10 distinct fields of view over two independent experiments. **f, g** Immunofluorescence detection of E-cadherin (E-cad), β-catenin (β-cat), claudin-1, and PARD3 in HaCaT cells treated with scrambled (Scr) or *hPIP5K1A*-targeting (PIP5K1A#1 and #2) siRNAs **f**. Images were taken from different fields of view. Quantification of the PM signals of E-cadherin, β-catenin, claudin-1, and PARD3 **g**. In total, 102 Scr, 104 PIP5K1A#1, and 112 PIP5K1A#2 cells were examined over two independent experiments for E-cadherin. In total, 117 Scr, 108 PIP5K1A#1, and 105 PIP5K1A#2 cells were examined over two independent experiments for β-catenin. In total, 115 Scr, 108 PIP5K1A#1, and 106 PIP5K1A#2 cells were examined over two independent experiments for claudin-1. In total, 102 Scr, 116 PIP5K1A#1, and 112 PIP5K1A#2 cells were examined over two independent experiments for PARD3 **g**. **h** mRNA expression level of *CDH2*. $N = 3$ for each group. ***$p = 0.0008$ (PIP5K1A#1). (versus HaCaT cells treated with scrambled siRNA). Data are represented as mean ± SD **a, b, h**. The box plots are presented with the elements: center line, median; box limits, Q1 and Q3; whiskers, 1.5× interquartile range. Outliers are also shown. Individual data points are displayed [gray points, data from the first experiments; black points, data from the second experiments **d, e, g**]. Significance was tested using one-way ANOVA with Tukey-Kramer's post hoc test **b, d, e, g, h** and the two-sided Welch's t-test **a**. ****$p < 0.0001$, *$p = 0.0413$ [versus control cells **a** and HaCaT cells treated with scrambled siRNA **b, d, e, g, h**]. Scale bar = 20 μm **c, f**. Source data are provided as a Source Data file.

migration in osteosarcoma cell lines. We performed a migration assay using a silicone insert with a defined cell-free gap. A total of 10 h after removal of the silicone insert, approximately 90% of the cell-free area was covered by migrated GFP- or Lyn-mPIP5Kmt-GFP-expressing U2OS cells, whereas Lyn-mPIP5Kwt-GFP-expressing U2OS cells covered less than half of the cell-free area (Supplementary Fig. 7c). Expression of Lyn-mPIP5Kwt-GFP also inhibited cell migration in the osteosarcoma cell line MG-63 (Supplementary Fig. 7d). These results indicate that elevation of PI(4,5)P$_2$ levels suppressed the migration of osteosarcoma cells. In addition, the Matrigel invasion assay revealed that over-expression of Lyn-mPIP5Kwt-GFP suppressed cell invasion in U2OS cells (Supplementary Fig. 7e). We further examined the effects of Lyn-mPIP5Kwt-GFP overexpression on cell proliferation. Lyn-mPIP5Kwt-GFP-expressing U2OS and MG-63 cells showed lower proliferative and clonogenic abilities than GFP- or Lyn-mPIP5Kmt-GFP-expressing cells (Supplementary Fig. 7f–h). Collectively, these results strongly suggest that elevation of PI(4,5)P$_2$ levels and acquisition of epithelial characteristics suppressed aggressive cellular phenotypes of osteosarcoma cells in vitro.

## Discussion
In this study, we found that epithelial cells had a higher amount of PI(4,5)P$_2$ than non-epithelial cells (Fig. 1a–c and Supplementary Fig. 1a). The intensity of PI(4,5)P$_2$ immunofluorescence signal was much stronger in the PM of epithelial cells than in non-epithelial cells. However, the total cellular PIP$_2$ amount was less than two-fold higher in epithelial cells than in non-epithelial cells, based on quantification using LC-MS/MS (Fig. 1h). Since PI(4,5)P$_2$ is suggested to be present not only in the PM but also in the inner cell membrane[26], the anti-PI(4,5)P$_2$ antibody may preferably recognize PI(4,5)P$_2$ in the PM, and the abundance of PM PI(4,5)P$_2$ may be selectively high in epithelial cells.

The intensity of PM cholesterol staining was stronger in the epithelial cells than in the non-epithelial cells (Fig. 5d, e, Supplementary Fig. 5c, d), suggesting the enrichment of cholesterol-rich and detergent-resistant lipid rafts in the PM of epithelial cells. The loss of epithelial characteristics is known to increase PM fluidity and destabilize lipid rafts[27]. Given that phospholipids with high lipid saturation status of acyl chains accumulate in lipid rafts[28], enrichment of PI(4,5)P$_2$ with high lipid saturation status in epithelial cells may contribute to PI(4,5)P$_2$ incorporation into lipid rafts in this cell type.

PI(4,5)P$_2$ depletion decreased PM cholesterol levels (Fig. 5a, b, Supplementary Fig. 5a, b); however, the mechanisms underlying this effect remain to be clarified. A previous report showed that the loss of adherens junctions decreases PM cholesterol levels[20]. Our results showed that PI(4,5)P$_2$ is required for normal formation of adherens junctions. Therefore, PI(4,5)P$_2$ may regulate PM cholesterol levels by inducing the formation of adherens junctions. Additionally, our results showed that PM cholesterol depletion impaired adherens junctions (Fig. 5h, i), suggesting that PM accumulation of cholesterol and formation of cell-cell junctions may be interdependent processes. OSBPL2 might also be involved in the regulation of PM cholesterol by PI(4,5)P$_2$. OSBPL2 delivers cholesterol to the PM in exchange for PI(4,5)P$_2$, and the PI(4,5)P$_2$ gradient between the PM and other organelles is reported to be the driving force for the OSBPL2-mediated targeted delivery of cholesterol to the PM[21]. Therefore, PI(4,5)P$_2$ depletion in the PM may diminish the PI(4,5)P$_2$ gradient, resulting in the inhibition of cholesterol transport by OSBPL2, thereby ultimately leading to a reduction of PM cholesterol levels.

PI(4,5)P$_2$ depletion impaired the PM localization of PARD3 and induced the loss of epithelial characteristics (Figs. 2 and 3). In line with our results, PI(4,5)P$_2$ has been reported to control the transport of Baz/Par-3 to the apical PM, and PI(4,5)P$_2$ depletion or mislocalization of Baz/Par-3 induces EMT-like phenotypes in Drosophila follicular epithelium[29]. PARD3 functions as an exocyst receptor[25], and vesicle exocytosis is known to occur at sites marked by high PI(4,5)P$_2$ concentration[30]. Furthermore, we identified PM t-SNAREs such as Syntaxin3, Syntaxin4, SNAP23, and SNAP29 as PI(4,5)P$_2$ proximal proteins in our experiment (Supplementary Data 1). Based on these reports and our own results, we propose that PARD3 is recruited to the PM via interaction with PI(4,5)P$_2$ and that it subsequently functions as the PM receptor for exocyst on E-cadherin-containing vesicles to establish adherens junctions and maintain or determine epithelial characteristics (Supplementary Fig. 8). A previous report showed that PIPKIγ interacts with E-cadherin and the exocyst component Exo70, generating PI(4,5)P$_2$ pools at the nascent adherens junctions[31]. According to the results of this study, PI(4,5)P$_2$- and PARD3-mediated delivery of E-cadherin-containing vesicles would be reinforced by a positive feedback loop, in which local PI(4,5)P$_2$ production by PIPKIγ further induces the recruitment of PARD3 and vesicles, enhancing the delivery of E-cadherin to establish stable adherens junctions.

In U2OS cells, elevation of PI(4,5)P$_2$ levels recruited N-cadherin to cell-cell contact and induced the development of cell-cell adhesion (Fig. 7e, f). PARD3 was recruited to the PM by PI(4,5)P$_2$ elevation (Fig. 7g, h), and expression of Lyn-hPARD3

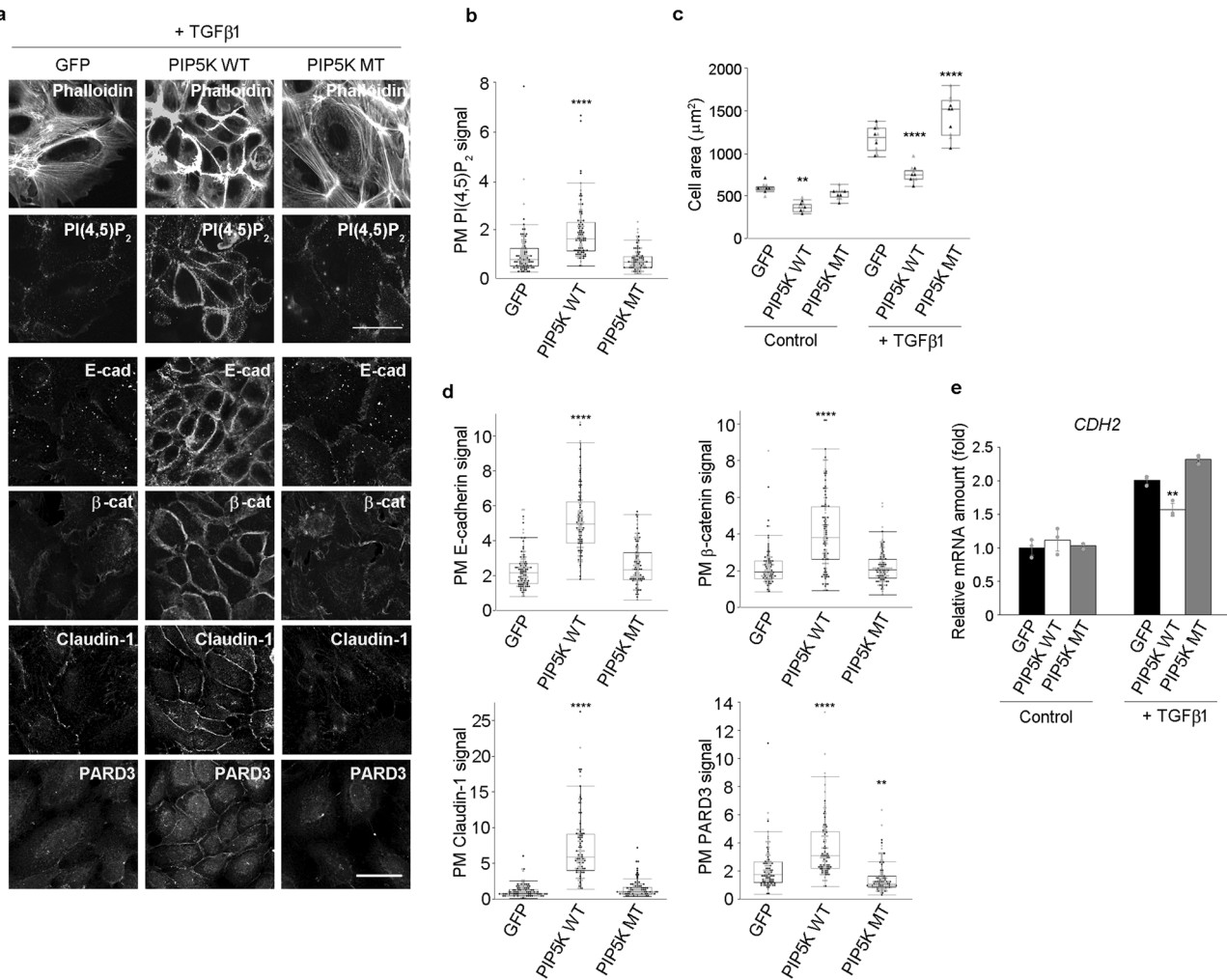

**Fig. 4 Elevation of PI(4,5)P$_2$ level partly suppresses TGFβ1-induced loss of epithelial characteristics. a–d** Detection of F-actin (phalloidin), PI(4,5)P$_2$, E-cadherin (E-cad), β-catenin (β-cat), claudin-1, and PARD3 in GFP-, Lyn-mPIP5Kwt-GFP (PIP5K WT)-, and Lyn-mPIP5Kmt-GFP (PIP5K MT)-expressing HaCaT cells treated with TGFβ1 for 72 h **a**. Images of E-cadherin, β-catenin, claudin-1, and PARD3 are taken from different fields of view. Quantification of the apicolateral plasma membrane (PM) PI(4,5)P$_2$ signals **b**; cell area **c**; PM signals of E-cadherin, β-catenin, claudin-1, and PARD3 **d**. In total, 109 GFP-, 100 PIP5K WT-, and 100 PIP5K MT-expressing cells were examined over two independent experiments **b**. The average cell area was calculated by analyzing either 10 distinct fields of view for control PIP5K WT-, TGFβ1-treated GFP-, PIP5K WT-, and PIP5K MT-expressing cells or 11 distinct fields of view for control GFP- and PIP5K MT-expressing cells over two independent experiments **c**. **p = 0.0017 (versus GFP-expressing cells) **c**. In total, 116 GFP-, 113 PIP5K WT-, and 108 PIP5K MT-expressing cells were examined over two independent experiments for E-cadherin. In total, 103 GFP-, 111 PIP5K WT-, and 112 PIP5K MT-expressing cells were examined over two independent experiments for β-catenin. In total, 112 GFP-, 112 PIP5K WT-, and 105 PIP5K MT-expressing cells were examined over two independent experiments for claudin-1. In total, 103 GFP-, 112 PIP5K WT-, and 110 PIP5K MT-expressing cells were examined over two independent experiments for PARD3 **d**. **p = 0.0046 (versus GFP-expressing cells) **d**. The box plots are presented with the elements: center line, median; box limits, Q1 and Q3; whiskers, 1.5× interquartile range. Outliers are also shown. **e** mRNA expression level of *CDH2*. N = 3 for each group. **p = 0.0066 **e** (versus TGFβ1-treated GFP-expressing cells). Data are represented as mean ± SD **e**. Individual data points are displayed [gray points, data from the first experiments; black points, data from the second experiments **b**, **c**, **d**]. Significance was tested using one-way ANOVA with Tukey-Kramer's post hoc test. ****p < 0.0001 (versus GFP-expressing cells). Scale bar = 30 μm **a**. Source data are provided as a Source Data file.

(710–1089)-V5 led to a phenotype similar to that achieved by PI(4,5)P$_2$ elevation in U2OS cells (Fig. 7j–l). Based on these observations, we suggest that PI(4,5)P$_2$ generated by Lyn-mPIP5Kwt-GFP recruits PARD3, which functions as a receptor for exocyst on N-cadherin-containing vesicles, resulting in the development of cell-cell adhesion structures in U2OS cells.

PIPs play a critical role in apical-basal polarization, which is an epithelial characteristic[32–35]. PI(4,5)P$_2$ is apically enriched and has been proposed as an apical identity determinant. In the polarized secretory epithelium of the salivary gland of the Drosophila larva, the protein Crumbs regulates the apical level of PI(4,5)P$_2$ by regulating Pten and Ocrl localization[36]. Given that PI(4,5)P$_2$ regulates vesicle exocytosis[30], polarized vesicle transport

controlled by PI(4,5)P$_2$ may play a critical role in the determination of epithelial characteristics.

Because PIPs are dynamically interconverted, changes in the amount of PI(4,5)P$_2$ may affect the levels of other PIPs. Therefore, we cannot exclude the possibility that changes in the levels of other PIPs also contribute to the determination of epithelial characteristics. Our results showed that PI(4,5)P$_2$ depletion resulted in the mislocalization of PARD3 (Figs. 2d, e, 3f, g) and that PARD3 downregulation induced the loss of epithelial characteristics (Fig. 6). Although other PIPs may also be involved in the determination of epithelial characteristics, PI(4,5)P$_2$ plays a critical role in the determination of epithelial characteristics by regulating the PM localization of PARD3.

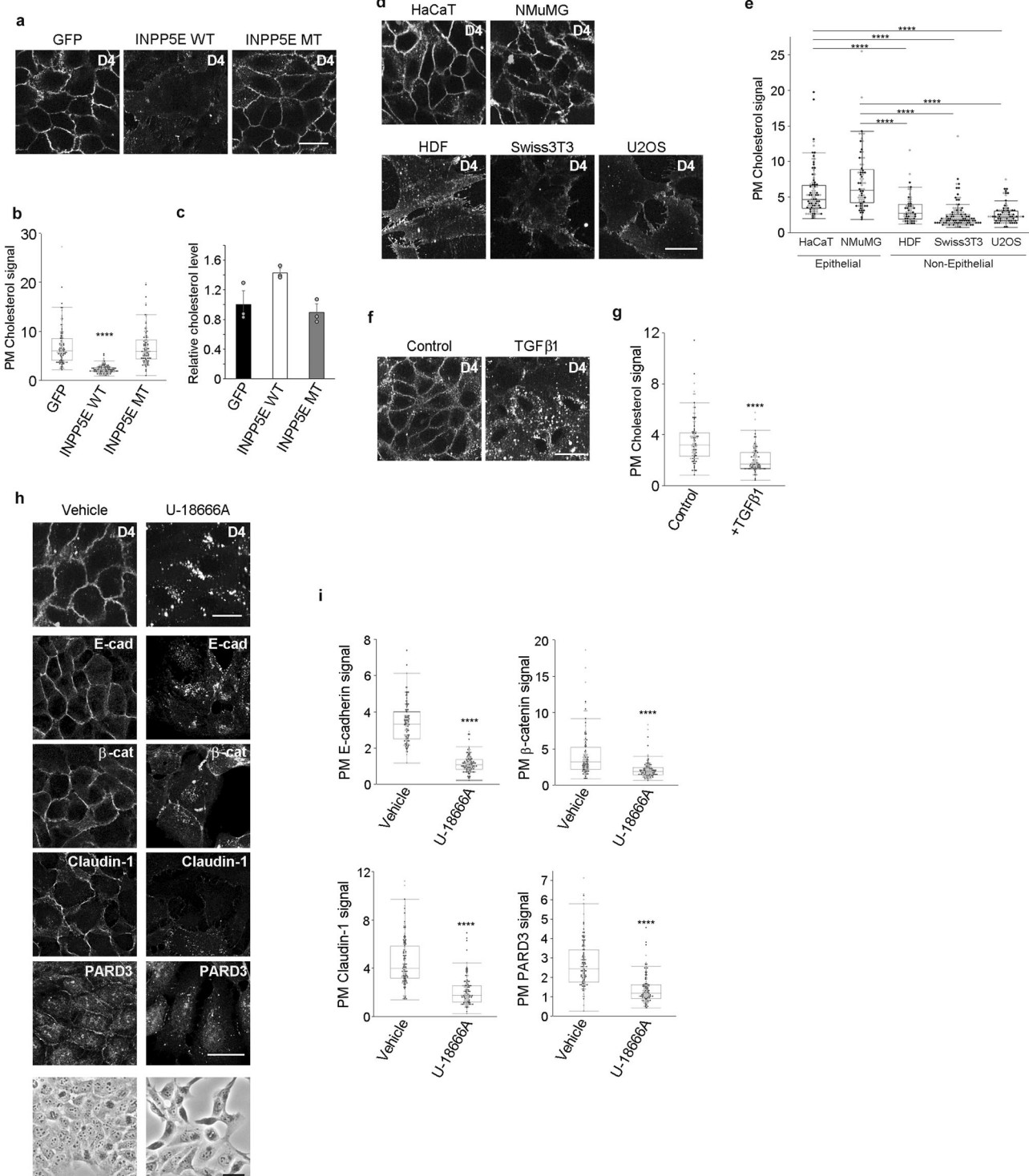

PLC activation is involved in the loss of epithelial characteristics[37]. Although PLC-generating second messengers and activation of their downstream signals are suggested to be the mechanism for PLC-mediated loss of epithelial characteristics, depletion of PI(4,5)P$_2$ by PLC may also contribute to it.

Although cells undergo complete EMT during development, tumor cells express epithelial and mesenchymal proteins simultaneously, which is termed as partial EMT[38]. Since PI(4,5)P$_2$-depleted epithelial cells retained expression of the epithelial-enriched gene *CDH1*, along with the induction of mesenchymal-enriched genes (Fig. 2g and Supplementary Fig. 2b), PI(4,5)P$_2$

depletion seemed to induce partial EMT. In addition to high invasive and metastatic activities, the onset of mesenchymal programs in cancer cells promotes their resistance to chemotherapies, promotes stem cell-like properties, prevents apoptosis and senescence, and contributes to immunosuppression[5]. Therefore, PI(4,5)P$_2$ may serve as a therapeutic target for tumor treatment by inhibiting partial EMT.

The role of transcription factors in the regulation of cell-type-specific transcriptional programs and the determination of the characteristics of epithelial and mesenchymal cells are well studied[39]. Although it has recently been suggested that

**Fig. 5 Depletion of PI(4,5)P$_2$ decreases the plasma membrane cholesterol. a, b** Detection of cholesterol using recombinant mCherry-D4 protein (D4) in GFP-, Lyn-INPP5Ewt-GFP (INPP5E WT)-, and Lyn-INPP5Emt-GFP (INPP5E MT)-expressing HaCaT cells **a**. Quantification of the plasma membrane (PM) cholesterol abundance **b**. In total, 103 GFP-, 119 INPP5E WT-, and 104 INPP5E MT-expressing cells were examined over two independent experiments **b**. **c** Relative cholesterol levels measured using cholesterol oxidase. Data are represented as mean ± SD **c**. $N = 3$ for each group. **d, e** Detection of cholesterol (D4) in epithelial (HaCaT and NMuMG) and non-epithelial (HDF, Swiss3T3, and U2OS) cell lines **d**. Quantification of PM cholesterol signals **e**. In total, 92 HaCaT, 71 NMuMG, 60 HDF, 121 Swiss3T3, 84 U2OS cells were examined over two independent experiments **e**. **f, g** Detection of cholesterol (D4) in HaCaT cells treated with or without TGFβ1 for 72 h **f**. Quantification of PM cholesterol signals **g**. In total, 100 control and 110 TGFβ1-treated cells were examined over two independent experiments **g**. **h, i** Detection of cholesterol (D4), E-cadherin (E-cad), β-catenin (β-cat), claudin-1, and PARD3 in untreated (Vehicle) or U-18666A-treated HaCaT cells **h**. Phase images are shown (bottom panels). Images were taken from different fields of view. Quantification of E-cadherin, β-catenin, claudin-1, and PARD3 in the PM **i**. In total, 115 vehicle- and 121 U-18666A-treated cells were examined over two independent experiments for E-cadherin. In total, 110 vehicle- and 120 U-18666A-treated cells were examined over two independent experiments for β-catenin. In total, 115 vehicle- and 120 U-18666A-treated cells were examined over two independent experiments for claudin-1. In total, 115 vehicle- and 120 U-18666A-treated cells were examined over two independent experiments for PARD3 **i**. The box plots are presented with the elements: center line, median; box limits, Q1 and Q3; whiskers, 1.5× interquartile range. Outliers are also shown. Individual data points are displayed (gray points, data from the first experiments; black points, data from the second experiments **b**, **e**, **g**, **i**). Significance was tested using one-way ANOVA with Tukey-Kramer's post hoc test **b**, **e** and the two-sided Welch's t-test **g**, **i**. ****$p < 0.0001$ (versus GFP-expressing cells **b**, HaCaT cells or NMuMG cells **e**, untreated cells **g**, **i**). Scale bar = 30 µm (**a**, **d**, **f**, **h** except for the bottom panels of **h**) or 50 µm (bottom panels of **h**). Source data are provided as a Source Data file.

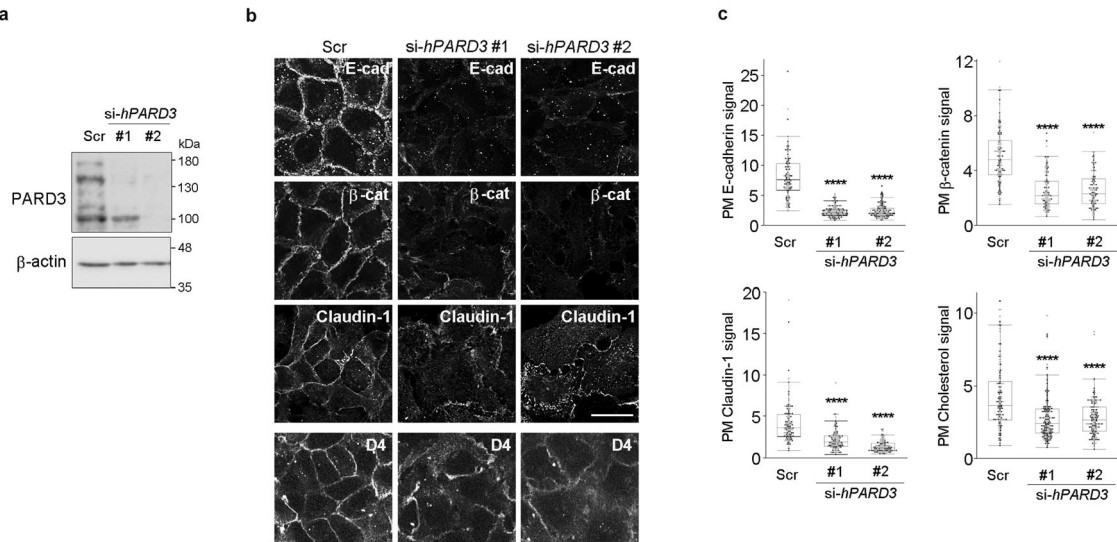

**Fig. 6 Plasma membrane PARD3 regulates epithelial characteristics. a** Immunoblotting for PARD3 in HaCaT cells treated with scrambled (Scr) or *PARD3*-targeting (PARD3#1 and #2) siRNAs. 180 kDa, 150 kDa, and 100 KDa forms of PARD3 were detected. β-actin was used as the loading control. Immunoblot data shown were representative of two independent experiments with similar results. **b, c** Detection of E-cadherin (E-cad), β-catenin (β-cat), claudin-1, and cholesterol (D4) in HaCaT cells treated with scrambled (Scr) or *PARD3*-targeting (PARD3#1 and #2) siRNAs **b**. Images were taken from different fields of view. Quantification of the plasma membrane (PM) signals of E-cadherin, β-catenin, claudin-1, and cholesterol **c**. In total, 103 Scr, 119 PARD3#1, and 111 PARD3#2 cells were examined over two independent experiments for E-cadherin. In total, 125 Scr, 104 PARD3#1, and 107 PARD3#2 cells were examined over two independent experiments for β-catenin. In total, 113 Scr, 113 PARD3#1, and 102 PARD3#2 cells were examined over two independent experiments for claudin-1. In total, 114 Scr, 152 PARD3#1, and 136 PARD3#2 cells were examined over two independent experiments for cholesterol **c**. The box plots are presented with the elements: center line, median; box limits, Q1 and Q3; whiskers, 1.5× interquartile range. Outliers are also shown. Individual data points are displayed (gray points, data from the first experiments; black points, data from the second experiments). Significance was tested using one-way ANOVA with Tukey-Kramer's post hoc test. ****$p < 0.0001$ (versus HaCaT cells treated with scrambled siRNA). Scale bar = 30 µm **b**. Source data are provided as a Source Data file.

phosphatidylethanolamine contributes to somatic cell reprogramming by regulating mesenchymal-epithelial transition[40], the role of lipids in the determination of cell characteristics is largely unknown. Our study revealed a link between PM PI(4,5)P$_2$ and the determination of epithelial characteristics.

## Methods

**Mice**. C57BL/6 J mice were purchased from CLEA Japan (Tokyo, Japan). Mice were housed in specific-pathogen-free conditions under a regular 12 h dark/light cycle at 23 ± 2 °C and 50% humidity, with food and water *ad libitum*. The skin of two newborn mice were harvested for the preparation of frozen sections. All animal experiments were approved by the Animal Experiments Review Board of the Tokyo University of Pharmacy and Life Sciences.

**Cell culture**. HaCaT cells (Cell Lines Service, Eppelheim, Germany) were cultured in Dulbecco's modified Eagle medium (DMEM, high glucose; Wako, Osaka, Japan) supplemented with 10% fetal bovine serum (FBS), 100 U/mL penicillin, and 100 µg/mL streptomycin. NMuMG cells (ATCC, Manassas, VA, USA) were cultured in DMEM (high glucose) supplemented with 10% FBS, 100 U/mL penicillin, 100 µg/mL streptomycin, and 10 µg/mL insulin (Sigma-Aldrich, St Louis, MO, USA). HDF (Kurabo, Osaka, Japan), Swiss3T3 (JCRB, Osaka, Japan), U2OS (ATCC), and MG-63 (JCRB) cells were cultured in DMEM (Nissui Pharmaceutical, Tokyo, Japan) supplemented with 10% FBS, 100 U/mL penicillin, and 100 µg/mL streptomycin. Cells were maintained in 5% CO$_2$ at 37 °C. TGFβ1 (PeproTech, Rocky Hill, NJ, USA) was used at a concentration of 50 ng/mL and treated for 72 h before sample collection, unless otherwise indicated. LY294002 (Cell Signaling Technology, Boston, MA, USA), Akt inhibitor VIII (Calbiochem, San Diego, CA, USA), and U73122 (Merck Millipore, Billerica, MA, USA) were administered at the indicated concentrations for 6, 1, and 3 h before sample collection, respectively. U-18666A (Abcam, Cambridge, MA, USA) was administered at a concentration of 1.25 µM

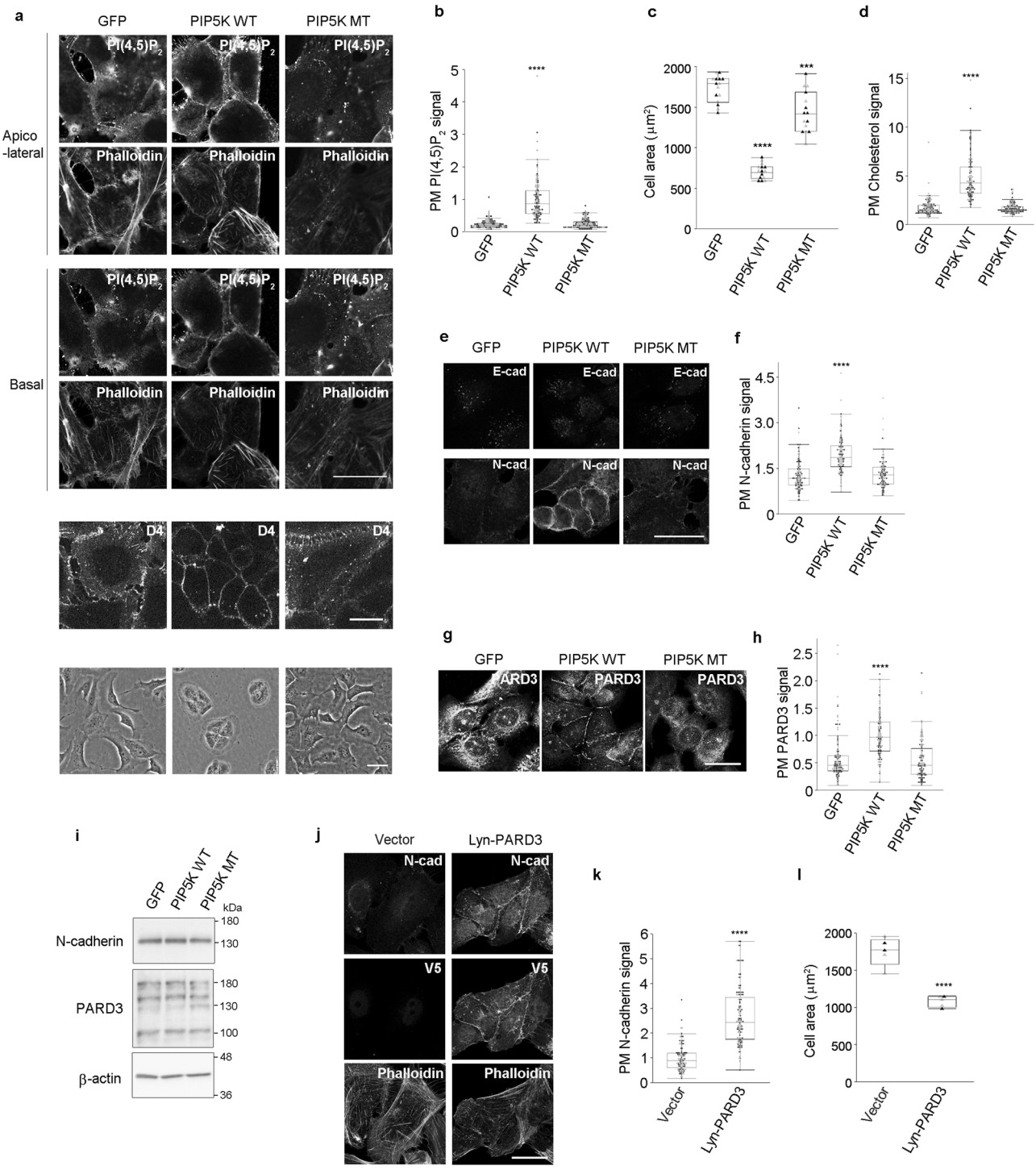

for 48 h before sample collection. Human *PIP5K1A*-targeting, human *OSBPL2*-targeting, human *PARD3*-targeting, and human *PLCδ1*-targeting siRNA duplexes were purchased from Thermo Fisher Scientific (Waltham, MA, USA). The siRNAs were transfected into HaCaT cells using Lipofectamine RNAiMax (Thermo Fisher Scientific), and the cells were analyzed after 72 h of incubation. For the transfection of human *PARD3*-targeting siRNA, cells were grown in the presence of the pan-caspase inhibitor Z-VAD-FMK (2 µM) (Adooq Bioscience, Irvine, CA, USA).

**Quantification of phospholipid levels**. Lipidome analysis was conducted according to the Lipidome lab Multiphospholipid Scan package (Lipidome lab, Akita, Japan), using liquid chromatography triple quadrupole mass spectrometry (LC-TQMS) based on the methods described previously[41, 42]. Briefly, total lipids were extracted from the samples using the Bligh-Dyer method[43]. An aliquot of the lower/organic phase was evaporated to dryness under $N_2$, and the residue was

dissolved in methanol for LC/MS/MS measurements of PC. PIPs were measured as described previously[44]. To analyze acidic phospholipids, such as PI and PIP, another aliquot of the same lipid extract was added to an equal volume of methanol before being loaded onto a diethylaminoethyl-cellulose column (Santa Cruz Biotechnology, Santa Cruz, CA, USA) pre-equilibrated with chloroform. After successive washes with chloroform/methanol (1:1, v/v), the acidic phospholipids were eluted with chloroform/methanol/HCl/water (12:12:1:1, v/v), followed by evaporation to dryness to give a residue, which was resolved in methanol. The resultant fraction was subjected to methylation with TMS-diazomethane before LC/MS/MS analysis[45]. LC-MS/MS analysis was performed using an UltiMate 3000 LC system (Thermo Fisher Scientific) equipped with an HTC PAL autosampler (CTC Analytics, Zwingen, Switzerland). The lipids were separated on a Waters X-Bridge C18 column (3.5 µm, 150 mm × 1.0 mm internal diameter) at room temperature (25 °C) using a gradient solvent system as follows: mobile phase A was isopropanol/methanol/water (5:1:4 v/v/v) supplemented with 5 mM ammonium

**Fig. 7 Elevation of PI(4,5)P₂ level confers epithelial characteristics to U2OS cells. a–f** Detection of PI(4,5)P₂ **a**, F-actin (phalloidin) **a**, cholesterol (D4) **a**, E-cadherin (E-cad) **e**, and N-cadherin (N-cad) **e**. Phase images are shown **a**. Quantification of PM PI(4,5)P₂ signals **b**, cell areas **c**, plasma membrane (PM) cholesterol signals **d**, and PM N-cadherin signals **f** of GFP-, Lyn-mPIP5Kwt-GFP (PIP5K WT)-, and Lyn-mPIP5Kmt-GFP (PIP5K MT)-expressing U2OS cells. In total, 107 GFP-, 101 PIP5K WT-, and 112 PIP5K MT-expressing cells were examined over two independent experiments **b**. The average cell area was calculated by analyzing 15 distinct fields of view over two independent experiments **c**. ***p = 0.0003 (versus GFP-expressing cells) **c**. In total, 111 GFP-, 102 PIP5K WT-, and 107 PIP5K MT-expressing cells were examined over two independent experiments **d**. In total, 112 GFP-, 108 PIP5K WT-, and 105 PIP5K MT-expressing cells were examined over two independent experiments **f**. **g, h** Detection of PARD3 in GFP-, PIP5K WT-, and PIP5K MT-expressing U2OS cells **g**. Quantification of PM PARD3 signals **h**. In total, 110 GFP-, 108 PIP5K WT-, and 105 PIP5K MT-expressing cells were examined over two independent experiments **h**. **i** Immunoblotting of N-cadherin and PARD3 in GFP-, PIP5K WT-, and PIP5K MT-expressing U2OS cells. 180 kDa, 150 kDa, and 100 KDa forms of PARD3 were detected. β-actin was used as the loading control. Immunoblot data shown are representative of two independent experiments with similar results. **j, k** Detection of N-cadherin (N-cad), Lyn-hPARD3 (710–1089)-V5 (V5), and F-actin (phalloidin) in U2OS cells with or without Lyn-hPARD3 (710–1089)-V5 (Vector and Lyn-PARD3, respectively) **j**. Quantification of N-cadherin PM signals **k**. In total, 107 vector- and 105 Lyn-PARD3-expressing cells were examined over two independent experiments **k**. **l** Cell areas of U2OS cells with or without Lyn-hPARD3 (710–1089)-V5 (Vector and Lyn-PARD3, respectively). The average cell area was calculated by analyzing five distinct fields of view over two independent experiments **l**. The box plots are presented with the elements: center line, median; box limits, Q1 and Q3; whiskers, 1.5× interquartile range. Outliers are also shown. Individual data points are displayed (gray points, data from the first experiments; black points, data from the second experiments). Significance was tested using one-way ANOVA with Tukey-Kramer's post hoc test **b, c, d, f, h** and the two-sided Welch's t-test **k, l**. ****p < 0.0001 (versus GFP-expressing cells or cells with empty vector). Scale bar = 30 μm (except for the bottom panels of **a**) or 50 μm (bottom panels of **a**). Source data are provided as a Source Data file.

formate and 0.05% ammonium hydroxide (28% in water); mobile phase B was isopropanol supplemented with 5 mM ammonium formate and 0.05% ammonium hydroxide (28% in water). The ratios of mobile phase A/B were 60%/40% (0 min), 40%/60% (0–1 min), 20%/80% (1–9 min), 5%/95% (9–11 min), 5%/95% (11–30 min), 95%/5% (30–31 min), 95%/5% (31–35 min), and 60%/40% (35–45 min). Flow rate was 25 μL/min. Lipid species were measured using selected reaction monitoring in positive ion mode with a triple-stage quadrupole mass spectrometer (TSQ Vantage AM, Thermo Fisher Scientific). The characteristic fragments of individual lipids were detected using a product ion scan (MS/MS mode). The PIPs species with different acyl chains analyzed were as follows: C32:0; C32:1; C32:2; C34:0; C34:1; C34:2; C36:0; C36:1; C36:2, C36:3; C36:4; C36:5; C38:0; C38:1; C38:2; C38:3; C38:4; C38:5; C38:6; C38:7; C40:0; C40:1; C40:2; C340:3; C40:4; C40;5; C40:6; C40:7; C42:0; C42;1; C42;2; C42:6; C42:7; C44:0; C44;1; C44;2; C44:6; C44:7; C44:12. Peak areas of individual species were normalized with those of the internal/surrogate standards 15:0-18:1-d7-PC, 15:0-18:1-d7-PI, 17:0-20:4 PI(4)P, 17:0-20:4 PI(4,5)P₂ (Avanti Polar Lipids, Alabaster, AL, USA), which were added to the samples before lipid extraction.

**Measurement of PI, PI(4)P, and PI(4,5)P₂.** HaCaT cell lipids were extracted by a modified Bligh-Dyer method and subjected to open column chromatography on DEAE Sepharose Fast Flow (GE Healthcare, Piscataway, NJ). After derivatization by trimethylsilyl diazomethane, PRMC-MS (phosphoinositide regioisomer measurement by chiral chromatography and mass spectrometry) was performed as described previously[19]. Supplementary Data 2 lists the pairs of *m/z* values of the precursor ions and the fragment ions used to detect each phosphoinositide species. The surrogate internal standard chemicals (C37:4 PI, C37:4 PI(4)P, C37:4 PI(4,5) P₂) were also monitored to determine the endogenous concentrations. Analyst 1.6.3 and MultiQuant (SCIEX) were used for data acquisition and data evaluation for peak integration, respectively.

**Retroviral expression.** pMXs-IP-Lyn-INPP5Ewt-GFP, pMXs-IP-Lyn-INPP5Emt-GFP, pMXs-IP-Lyn-mPIP5Kwt-GFP, pMXs-IP-Lyn-mPIP5Kmt-GFP, and pMXs-IP-Lyn-hPARD3 (710–1089)-V5 were constructed by the fusion of the PM-targeting signal (MGCIKSKRKD, a myristoylation motif derived from Lyn tyrosine kinase) to the N-terminus of the catalytic domain (residues 214–644) of human INPP5E-GFP, D477N mutant of the catalytic domain of human INPP5E-GFP, full-length mouse PIP5K1C635-GFP, D277A mutant of full-length mouse PIP5K1C635-GFP, and exocyst-docking region of human PARD3-V5 (residues 710–1089), followed by subcloning into pMXs-IP retroviral vectors. Human SNAI1 and SNAI2 were subcloned into a pMXs-IP retroviral vector modified by introducing an HA tag to the C-terminus. GFP C-terminally fused to full-length human PIP5K1A, siRNA-resistant full-length human PIP5K1A, GFP N-terminally fused to the wild-type PH domain of human PLCδ1, and V5 and APEX2 N-terminally fused to the wild-type, or R40L mutant PH domain of PLCδ1; these were subcloned into a pMXs-IP retroviral vector. Retroviruses were generated by transfecting retroviral plasmid DNA using Lipofectamine 2000 (Thermo Fisher Scientific) into Platinum-A cells. Retrovirus mixed with polybrene at a final concentration of 1 μg/mL (Santa Cruz Biotechnology) was used to infect the target cells. One day post-infection, the cells were switched to fresh culture medium containing 1 or 2 μg/mL puromycin (Sigma-Aldrich) for 72 h before the experiments.

**Immunofluorescence.** Immunofluorescence analysis of PI(4,5)P₂ in mouse skin was performed using frozen sections of skin of newborn mice with the TSA Plus Cyanine 3 System (PerkinElmer, Waltham, MA, USA). For the immuno-fluorescence analysis of cultured cells, the cells were stained with antibodies against E-cadherin (BD Biosciences, San Jose, CA, USA), CTNNB1 (β-catenin) (BioLegend, San Diego, CA, USA), CLDN1 (Claudin-1) (Santa Cruz Biotechnology), PARD3 (Merck Millipore), N-cadherin (BD Biosciences), V5 (BioLegend), phalloidin Alexa647 (Thermo Fisher Scientific), and phalloidin CF647 (Biotium, Hayward, CA, USA). PI(4,5)P₂ staining was performed as previously described[46]. Cholesterol was detected using a Cholesterol Assay Kit (Cell-Based) (Abcam). Samples were observed using a fluorescence microscope (BZ-X700 and BZ-X810; Keyence, Osaka, Japan) or a confocal laser microscope (FV1200; Olympus, Tokyo, Japan or LSM900; Zeiss, Jena, Germany). The apicolateral plane was located 1.3 μm above the basal plane for HaCaT cells. For U2OS cells, a plane located at 0.8 μm above the basal plane was used for the analysis. Images were quantified using the ImageJ software (NIH) and Zeiss ZEN imaging software (Zeiss). To quantify the PM signal of E-cadherin, β-catenin, claudin-1, PARD3, and filipin, the ratio between the fluorescence intensity measured in two rectangular regions of interest (ROIs) from the apicolateral PM region and two rectangular ROIs in the cytosol was calculated. For the PM PI(4,5)P₂ signal, the PI(4,5)P₂ fluorescence intensity measured in two ROIs from the apicolateral PM region was divided by the fluorescence intensity of the PM marker [Alexa Fluor 350 conjugate wheat germ agglutinin (WGA) (Thermo Fisher Scientific)], measured in the same ROIs, to normalize the surface area of the PM. At least 75 cells were analyzed from two independent experiments. For cell area analysis, the total cell area was divided by the number of cells per field of view to determine the average cell area. At least five fields of view and 100 cells were analyzed from two independent experiments.

**mCherry-D4 labeling of cholesterol.** Cholesterol detection using recombinant mCherry-D4 protein was performed as reported previously[47]. Briefly, cells were incubated with 5 μg/ml mCherry-D4 and Hoechst 33342 in binding buffer consisting of 0.1% bovine serum albumin/phosphate buffered saline (PBS) at room temperature to label the cholesterol in the outer leaflet of the PM. The cells were then washed and fixed with 4% paraformaldehyde in PBS. After fixation, the cells were washed and briefly submerged in liquid nitrogen to permeabilize the PM. The cells were then incubated with 5 μg/ml mCherry-D4 in binding buffer at room temperature to label the inner leaflet cholesterol of PM. After mounting with glycerol, the samples were observed using a fluorescence microscope (BZ-X810; Keyence). To quantify the PM signal of mCherry-D4, the ratio between the fluorescence intensity measured in two ROIs from the apicolateral PM region and two rectangular ROIs in the cytosol was calculated. At least 60 cells were analyzed from two independent experiments.

**Quantification of cholesterol and cholesterol ester levels.** Cholesterol and cholesterol ester levels were measured using a total cholesterol assay kit (Cell Biolabs, San Diego, CA, USA) following the manufacturer's instructions. All values were normalized to the cell number of each sample.

**Real-time quantitative PCR.** RNA isolation, cDNA synthesis, and real-time quantitative PCR were performed as described previously[14]. The primers used for real-time quantitative PCR are listed in Supplementary Table 1. The expression level of the genes of interest were normalized to those of the *ACTB* or *GAPDH*.

**Isolation and determination of PI(4,5)P$_2$ proximal proteins**. HaCaT cells expressing V5 and APEX2-fused wild-type and mutated (R40L) PH domain of PLCδ1 were labeled with light and heavy isotope amino acids, respectively, using the SILAC$^{TM}$ Protein Quantitation kit (Thermo Fisher Scientific) following manufacturer's instructions. Briefly, cells were grown in DMEM containing either unlabeled l-arginine and l-lysine (Arg$^0$, Lys$^0$) (light) or l-[$^{13}$C$_6$,$^{15}$N$_4$]-arginine and l-[$^{13}$C$_6$,$^{15}$N$_2$]-lysine (Arg$^{10}$ and Lys$^8$) (heavy) supplemented with 2 mM L-glutamine, 100 U/mL penicillin, 100 μg/mL streptomycin, and 10% dialyzed FBS. Cells were grown in SILAC medium for at least seven cell doublings. Labeled cells were incubated with 500 μM biotin-phenol (Iris Biotech GmbH, Marktredewitz, Germany) in SILAC medium for 30 min at 37 °C. Then, H$_2$O$_2$ was added to the cells at a final concentration of 1 mM for 1 min at room temperature, after which the cells were washed twice with the quencher solution (10 mM sodium ascorbate, 10 mM sodium azide, and 5 mM Trolox in PBS), twice with PBS, and once more with the quencher solution. The cells were then collected and lysed in radio-immunoprecipitation assay buffer. Three independent biotinylation experiments using V5-APEX2-fused PH domain- and R40L mutant-expressing HaCaT cells were performed. Then, the same amount of three light samples and three heavy samples were mixed and used for the isolation of biotinylated proteins with streptavidin magnetic beads (Thermo Fisher Scientific). The biotinylated proteins were then eluted by boiling the beads in a 3× protein loading buffer supplemented with 2 mM biotin and 20 mM dithiothreitol for 10 min. The biotinylated proteins were separated on a 10% Bis-Tris Novex mini-gel (Thermo Fisher Scientific) using the MES buffer system. The gel was stained with Coomassie and excised into 10 equally sized segments. Gel segments were processed using ProGest robot (DigiLab, Hopkinton, MA, USA) according to the following protocol. First, the gel segments were reduced with 10 mM dithiothreitol at 60 °C, followed by alkylation with 50 mM iodoacetamide at room temperature. Then, the segments were digested with trypsin (Promega, Madison, WI, USA) at 37 °C for 4 h and subsequently quenched with formic acid. The supernatant was analyzed directly without further processing using nano LC/MS/MS with a Waters NanoAcquity HPLC system interfaced with a ThermoFisher Q Exactive. Peptides were loaded onto a trapping column and eluted over a 75-mm analytical column at 350 nL/min; both columns were packed with Luna C18 resin (Phenomenex, Torrance, CA, USA). The mass spectrometer was operated in data-dependent mode, with MS and MS/MS performed in the Orbitrap at 70,000 and 17,500 full width at half maximum resolution, respectively. The 15 most abundant ions were selected for MS/MS analysis. Data were processed using MaxQuant software 1.6.2.3 (www.maxquant.org), which served the following functions: recalibration of the MS data, filtering of database search results at 1% protein and peptide false discovery rate (FDR), and calculation of SILAC heavy:light ratios. Data were searched using a local copy of Andromeda with the following parameters: enzyme, trypsin; database, Swissprot Human (concatenated forward and reverse plus common contaminants); fixed modification, carbamidomethyl (C); variable modifications, oxidation (M), acetyl (protein N-term), $^{13}$C$_6$/$^{15}$N$_2$ (K), and $^{13}$C$_6$/$^{15}$N$_4$ (R); and fragment mass tolerance, 20 ppm. Pertinent MaxQuant settings were as follows: peptide FDR, 0.01; protein FDR, 0.01; minimum peptide length, 7; minimum unique peptides, 0; minimum ratio count, 2; requantify, TRUE; and second peptide, TRUE. Proteins were identified as PI(4,5)P$_2$ proximal protein when the protein's heavy/light SILAC ratio was less than 0.2. Visualization of the biotinylated proteins was performed using Alexa Fluor 594 streptavidin (Thermo Fisher Scientific).

**Immunoblot analysis**. Cell lysates were subjected to sodium dodecyl sulfate–polyacrylamide gel electrophoresis and subsequently transferred onto polyvinylidene fluoride membranes. The membranes were blocked with 10% skim milk and incubated with the antibodies against E-cadherin (BioLegend), β-catenin (BioLegend), claudin-1 (Santa Cruz Biotechnology), PARD3 (Merck Millipore), Akt (Cell Signaling Technology), phospho-Akt (Ser473) (Cell Signaling Technology), PLCδ1 (Santa Cruz Biotechnology), N-cadherin (BD Biosciences), and β-actin (Sigma-Aldrich), followed by incubation with a horseradish peroxidase-conjugated secondary antibody (Dako, Glostrup, Denmark). Images were captured using LuminoGraph I (ATTO, Tokyo, Japan).

**Cell invasion and migration assays**. For the invasion and migration assays, BioCoat Matrigel Invasion Chambers (Corning, Corning, NY, USA) and Culture-Insert 2 Well (Ibidi GmbH, Gräfelfing, Germany), respectively, were used. In the invasion assay, cells were suspended in serum-free DMEM and seeded on top of the chamber's membrane, while DMEM supplemented with 10% FBS was placed at the bottom. Cells were incubated for 22 h, fixed in 4% paraformaldehyde, and stained with crystal violet. Invaded cells were imaged and counted using a BZ-X700 microscope (Keyence). For the migration assay, U2OS cells or MG-63 cells were seeded onto wells containing a silicon insert with a defined cell-free gap. U2OS cells were then cultured with DMEM supplemented with 10% FBS and 2 μg/mL mitomycin C for 16 h. MG-63 cells were cultured for 16 h, and 2 μg/mL mitomycin C was added to the DMEM supplemented with 10% FBS during the last 4 h of culture. The silicon insert was then removed, and the cell-free areas were imaged and measured using a BZ-X700 and BZ-X810 microscope (Keyence).

**Cell proliferation assay**. U2OS and MG-63 cells were seeded in 96-well plates (2,000 cells per well). At 24, 48, and 72 h, the cells were stained with crystal violet and cell proliferation was analyzed by measuring the optical density (OD, 595 nm) using a microplate reader (Molecular Devices, Sunnyvale, CA, USA).

**Colony formation assay**. U2OS cells were seeded onto 12-well plates (50 or 200 cells per well). After 7 days of incubation, the media was removed, and the cells were washed with PBS prior to the addition of 2% glutaraldehyde. The cells were then washed with PBS and incubated with 0.1% crystal violet. The stained cells were washed with PBS, and cell colonies were imaged and counted using a microscope (BZ-X700; Keyence).

**Statistical analysis**. All statistical analyses were performed in JMP Pro 14 (SAS Institute). The results were expressed as mean ± standard deviation (SD). The significance of differences was determined using two-sided Welch's t-test. One-way ANOVA with Tukey–Kramer method was used to adjust for multiple comparisons.

**Reporting summary**. Further information on research design is available in the Nature Research Reporting Summary linked to this article.

## Data availability

The mass spectrometry–based proteomics data generated in this study have been deposited in the ProteomeXchange Consortium via the PRIDE partner repository under accession code PXD031266. Source data are provided with this paper.

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

## Acknowledgements

We thank Drs. T. Miyamoto, J. Ikenouchi, Y. Fujikawa, and K. Sako for their helpful advice, and Messrs. R. Makino, N. Honda, R. Usami, A. Katayanagi, S. Suda, and Ms. A. Aoyagi for the technical help. This work was supported by a Grant-in-Aid for Scientific Research (B) 18H02575, the Takeda Science Foundation, the Sumitomo Foundation, the Terumo Life Science Foundation, the Mochida Memorial Foundation for Medical and Pharmaceutical Research, the Ichiro Kanehara Foundation, the Hamaguchi Foundation for the Advancement of Biochemistry, Nanken-Kyoten/Medical Research Center Initiative for High Depth Omics, TMDU and PRIME (JP17gm5910017) to Y.N., a Grant-in-Aid for Young Scientists 19K16080 to K.K., and a Grant-in-Aid for Scientific Research (B) 17H04051 to K.F.

## Author contributions

Y.N. conceived the project and wrote the manuscript with the support of K.F., K.K., M.K., K.Tsujita, and T.S. Y.N. designed the experiments. K.K., M.S., M.K., H.K., Y.S., Y.C., R.F., T.F., H.N., T.S., J.H., J.S., and Y.N. performed the experiments and analyzed the data. T.K., K.Tsujita, K.Tanaka. and T.I. provided key reagents. All the authors contributed to the discussion of the results.

## Competing interests

H.N. works for Lipidome Lab Co., Ltd. (experiments and data analysis, and writing-review and editing). The remaining authors declare no competing interests.
