## [Peer Review File · Nature Communications]

Plasma membrane phosphatidylinositol (4,5)-bisphosphate is critical for determination of epithelial characteristicsEditorial Note: Parts of this Peer Review File have been redacted as indicated to maintain the confidentiality of unpublished data.

Reviewers' Comments:

Reviewer #1:

Remarks to the Author:

In this manuscript, the authors used APEX fused PH domain of PLC δ 1 as bait, to probe proteins in proximity with PIP2. The experimental design and data interpretation are generally ok, however, the authors may provide additional information to further support and strengthen their results.

1. It would be great that the authors using imaging techniques to confirm the colocalization of PIP2 and the fused protein. Although we already know PIP2 will bind the PH domain of PLC δ 1, does the fusion with APEX affected the expression and localization of the PH domain? Dose the APEX fused R40L mutation really does not colocalize with PIP2?

2. The authors have provided great replication information of other experiments but not the IP-MS. How many IP-MS samples are prepared and analyzed? In the Methods section the authors defined PIP2 proximal protein as proteins with heavy/light SILAC ratio less than 0.2 but no statistical information here (p? FDR?), making people doubt the IP-MS was only performed once.

Reviewer #2:

Remarks to the Author:

Nature COMMS-21-16175 Kanemaru

In epithelial cells, it was known that PIP2 is enriched in the apical plasma membrane, though it was not known if such PIP2 played a role in differentiation into the epithelial phenotype. Here, the authors show that PIP2 is enriched in the plasma membrane of epithelial cells, depletion of plasma membrane PIP2 cause loss of the epithelial phenotype, and forced production of PIP2 promotes the epithelial phenotype. Biotin proximity labelling identified epithelial junction proteins and collagen 17 as near to PIP2 and regulated by this lipid. They conclude that PIP2 is an important determinant of the epithelial phenotype.

Overall, this paper is potentially appropriate for Nature Communications in terms of the importance of the work and the quality of the data and presentation. However, some additional data is needed to make it more convincing.

1. Can the authors please provide X-Z confocal images of the distribution of PIP2 and some of the other plasma membrane components?

2. In Figure 2, can they please show the localization of CDH1 under various conditions?

3. For all of the RNAi experiments, they need specificity controls. This is usually done by expression of an RNAi-resistant protein.

4. In quantitation of localization of PIP2 and other plasma membrane components (e.g. Figure 1), they need to account for folding of the plasma membrane and other things that affect the surface area of the plasma membrane, such as microvilli.

Reviewer #3:

Remarks to the Author:

This manuscript by Kanemaru et al., describes interesting phenotypes related to loss of PI(4,5)P2 from epithelial cell membranes. They first report that epidermal cells, and NMuMG mammary epithelial cells, express more PI(4,5)P2 than dermal fibroblasts and two other non-epithelial cell lines. Depleting PI(4,5)P2 by expression of a membrane-targeted phosphatase, Lyn10-INPP5Ewt-GFP, disrupted F-actin organization and increased cell area. Cells took elongated forms reminiscent of a mesenchymal phenotype. Although there was not a classical EMT, as marked by loss of CDH1, N-cadherin (CDH2) levels were increased. Similar phenotypes were caused by TGFbeta treatment, and by depletion of PIP5K1A, while over-expression of PIP5K1A partly suppressed the TGFbeta1-induced phenotype. The authors make an interesting observation that depletion of PI(4,5)P2 decreases plasma membrane cholesterol, and suggest that cholesterol reduction is the reason behind the morphological changes in the epithelial cells. [Redacted]

Overall, the authors have assembled an interesting collection of data, implicating PIP2 levels in the maintenance of the epithelial phenotype through maintenance of cholesterol levels [Redacted]. However, there are several issues that I feel weaken the overall impact of the study:

- 1) Only three assays address the change in phenotype – cell area (which is not very informative), F-actin distribution, and CDH2 expression. Near the end of the paper they also show a mislocalization of E-cadherin. There is no analysis of tight junctions or polarity protein localization, and no measurement of other mesenchymal markers such as vimentin. Given that this loss of the “epithelial phenotype” is central to the study, I feel that a more in-depth characterization is required.
- 2) The authors try to make general conclusions about the role of PIP2, cholesterol [Redacted]
- 3) Several of the conclusions depend on the use of a single small molecule inhibitor, and negative results have no controls to validate the effectiveness of inhibition. Knockdown of ORP2 could be used as an alternative to reduce PM cholesterol. Also, filipin binding as a measure of cholesterol can be problematic as the response is non-linear, and changes in lipid composition might cause filipin to bind to other lipid species.
- 4) The organization of the paper is not optimal – the effects of increasing PIP2 levels (Supplementary Figure 5) are only presented after the data about cholesterol levels (Figure 4), and the effects of changing PIP2 levels on E-cadherin localization are not shown until after the effect of cholesterol reduction on E-cadherin have been discussed. Figure 6, on the effects of PIP2 elevation on U2OS cells is not very interesting – if the authors want to argue that increased PIP2 will reduce tumorigenicity then they need to do in vivo assays to measure this. Effects on migration are not particularly novel and seem peripheral to the main arguments of the manuscript. I would suggest these data go into a Supplementary figure or be left out, and the effects of increased PIP2 on TGFb-induced phenotypes be moved to the main figures.
- 5) Most importantly, the mechanistic aspects of the manuscript are rather superficial – the authors do not investigate the mechanism that ensures PIP2 levels are higher in epithelia than in non-epithelial cells, except in response to TGFbeta. They do not explore how TGFbeta causes a decrease in PIP5K1A, or how reduced PIP2 causes a decrease in PM cholesterol [Redacted]
- 6) Finally, the authors do not cite multiple prior papers describing a role for PIP2 on epithelial polarity, including one showing that PI(4,5)P2 controls apical size by tethering Par3 (Claret Curr Biol 2014), another that Crumbs regulates apical levels of PI(4,5)P2 (Lattner ELife 2019), and one that vesicle exocytosis occurs at PI(4,5)P2 rich domains (Martin BBA 2015).
- 7) It seems likely, based on prior literature, that a key mechanism for the observed loss of epithelial characteristics is a loss of polarized vesicle transport (see for instance the Martin review), but this is not discussed at all.

Overall, my conclusion is that the study demonstrates an interesting phenotype caused by depletion of

PIP2, and begins to explore mechanism, but that the work is too superficial as presented to be suitable for publication by Nature Communications.

Responses to Reviewer 1

1. It would be great that the authors using imaging techniques to confirm the colocalization of PIP2 and the fused protein. Although we already know PIP2 will bind the PH domain of PLC δ 1, does the fusion with APEX affected the expression and localization of the PH domain? Dose the APEX fused R40L mutation really does not colocalize with PIP2?

We thank the reviewer for this suggestion. According to the reviewer's suggestion, we performed double immunofluorescence for PI(4,5)P₂ and APEX2-fused PH domains. Anti-PI(4,5)P₂ monoclonal antibody cannot detect PM PI(4,5)P₂ in cells expressing the APEX2-fused PH domain, probably because of PI(4,5)P₂ masking by APEX2-fused PH domain. Therefore, we performed single immunofluorescence of the APEX2-fused PH domain and its R40L mutant and confirmed that the APEX2-fused PH domain showed PM accumulation, whereas this was not observed with the R40L mutant (Supplementary Fig. 6a in the revised manuscript). This observation strongly suggests that fusion with APEX2 did not affect the expression and localization of the PH domain. We also examined the localization of biotinylated proteins using fluorescent streptavidin and confirmed that APEX2-fused PH domain biotinylated PM proteins, whereas its R40L mutant did not (Supplementary Fig. 6b in the revised manuscript).

2. The authors have provided great replication information of other experiments but not the IP-MS. How many IP-MS samples are prepared and analyzed? In the Methods section the authors defined PIP2 proximal protein as proteins with heavy/light SILAC ratio less than 0.2 but no statistical information here (p? FDR?), making people doubt the IP-MS was only performed once.

We apologize for the insufficient information on SILAC methods. The LC/MS/MS analysis was performed once. To prepare samples for LC/MS/MS analysis, we performed three independent biotinylation experiments using APEX2-fused PH domain- and R40L mutant-expressing cells. Then, the same amount of three heavy samples and three light samples from three independent biotinylation experiments were mixed and used for the isolation of biotinylated proteins with streptavidin magnetic beads. We have added a detailed description of the methods of SILAC in the revised manuscript (p.35 l.2-p. 37 l.12 in the revised manuscript).

Responses to Reviewer 2

1. Can the authors please provide X-Z confocal images of the distribution of PIP2 and some of the other plasma membrane components?

As the reviewer suggested, we obtained X-Z images of PI(4,5)P₂ and wheat germ agglutinin (WGA), which recognizes cell surface sugar chains. We confirmed that epithelial cells have a higher intensity of PM PI(4,5)P₂ signal than non-epithelial cells or epithelial cells undergoing epithelial-mesenchymal transition (Supplementary Fig. 1a, d, f in the revised manuscript). We also confirmed that the intensity of the PM PI(4,5)P₂ signal is decreased upon PI(4,5)P₂ depletion by expression of Lyn-INPP5Ewt-GFP (Supplementary Fig. 2a in the revised manuscript).

2. In Figure 2, can they please show the localization of CDH1 under various conditions?

We have added the data on localization of CDH1 (E-cadherin) in GFP-, Lyn-INPP5Ewt-GFP-, and Lyn-INPP5Ewt-GFP-expressing HaCaT cells to Fig. 2d-f in the revised manuscript. We also examined the localization of other junctional and polarity proteins and confirmed that PM localization of these proteins was reduced by PI(4,5)P₂ depletion (Fig. 2d-f in the revised manuscript).

3. For all of the RNAi experiments, they need specificity controls. This is usually done by expression of an RNAi-resistant protein.

We thank the reviewer for this suggestion. As the reviewer suggested, we generated an expression vector for siRNA-resistant hPIP5K1A, hORP2, and hPar3 by introducing synonymous substitutions into the siRNA-targeting sequences. We confirmed that siRNA-resistant hPIP5K1A-expressing cells maintained their epithelial characteristics even after the introduction of hPIP5K1A-targeting siRNA (Supplementary Fig. 3d-f in the revised manuscript).

siRNA-resistant Par3 (180 KDa) was not efficiently expressed in HaCaT cells, probably because of the higher molecular weight of Par3. Instead, we used two siRNAs recognizing distinct sequences of Par3 mRNA to minimize the risk of off-target effects of siRNA and confirmed that both siRNAs induced similar phenotypes (Fig. 6a-c in the revised manuscript).

Regarding ORP2, we performed retroviral expression of siRNA-resistant ORP2 and found that exogenous expression of ORP2 reduced PM PI(4,5)P₂, as reported previously (Wang *et al.*, *Mol. Cell* 2019 PMID: 30581148), resulting in the loss of epithelial morphology in HaCaT cells before introducing siRNA (please see attached data on this page). Therefore, we used two siRNAs recognizing distinct sequences of ORP2 mRNA and confirmed that both siRNAs induced similar phenotypes (Supplementary Fig. 5g-i in the revised manuscript).

4. In quantitation of localization of PIP2 and other plasma membrane components (e.g. Figure 1), they need to account for folding of the plasma membrane and other things that affect the surface area of the plasma membrane, such as microvilli.

We apologize for the insufficient explanation of the PI(4,5)P₂ quantification method. We used a fluorescent WGA conjugate that recognizes the cell surface sugar chain to label the plasma membrane. Since the intensity of the fluorescent WGA conjugate reflects the surface area of the plasma membrane, we used the intensity of the fluorescent WGA conjugate to normalize the intensity of the PI(4,5)P₂ signal. We have added a description to the revised manuscript (p.33 1.4-p.33 1.7 in the revised manuscript).

Responses to Reviewer 3

1) Only three assays address the change in phenotype - cell area (which is not very informative), F-actin distribution, and CDH2 expression. Near the end of the paper they also show a mislocalization of E-cadherin. There is no analysis of tight junctions or polarity protein localization, and no measurement of other mesenchymal markers such as vimentin. Given that this loss of the “epithelial phenotype” is central to the study, I feel that a more in-depth characterization is required.

We appreciate the reviewer’s comment. We examined the abundance of a tight junction protein, Claudin-1, and a polarity protein, Par3, in the PM and confirmed that their PM levels were decreased upon epithelial-mesenchymal transition induction (Fig. 4a, d and Supplementary Fig. 1d-g in the revised manuscript), PI(4,5)P₂ depletion (Fig. 2d,e and Fig. 3f, g in the revised manuscript), PM cholesterol depletion (Fig. 5h, i, and Supplementary Fig. 5h, i in the revised manuscript), and Par3 depletion (Fig. 6b, c in the revised manuscript). We also examined the expression level of vimentin and found that vimentin expression was not induced by PI(4,5)P₂ depletion (Supplementary Fig. 2b in the revised manuscript). This observation indicates that PI(4,5)P₂ depletion results in the loss of epithelial characteristics, but does not induce a complete gain of mesenchymal characteristics (p.9 1.11-13 in the revised manuscript).

2) The authors try to make general conclusions about the role of PIP2, cholesterol [redacted]

[redacted]

3) Several of the conclusions depend on the use of a single small molecule inhibitor, and negative results have no controls to validate the effectiveness of inhibition. Knockdown of ORP2 could be used as an alternative to reduce PM cholesterol. Also, filipin binding as a measure of cholesterol can be problematic as the response is non-linear, and changes in lipid composition might cause filipin to bind to other lipid species.

As suggested by the reviewer, we performed ORP2 knockdown to decrease PM cholesterol levels and

confirmed that ORP2 knockdown induces loss of epithelial characteristics in HaCaT cells (Supplementary Fig. 5g-i in the revised manuscript). For cholesterol detection, we used recombinant mCherry-D4 protein (Ishitsuka *et al.*, *J. Lipid Res.* 2011 PMID: 21862703) to visualize cholesterol in the cell membrane (Fig. 5a, d, f, h, Fig. 6b, Fig. 7a, and Supplementary Fig. 5h in the revised manuscript).

We also confirmed the effective inhibition of PI3K by LY294002 by examining the level of phosphorylated Akt (Supplementary Fig. 4a in the revised manuscript). We further inhibited Akt, one of the main downstream molecules of PI3K, using Akt inhibitor VIII and confirmed that Akt inhibition did not induce morphological changes that were observed in PI(4,5)P₂-depleted cells (Supplementary Fig. 4b in the revised manuscript). For PLC inhibition, it is difficult to examine the effectiveness of the inhibitor under normal culture conditions without PLC-activating stimulation. Therefore, we inhibited PLC activity in keratinocyte cell lines using an alternative method. Because PLCδ1 is responsible for most PLC activity in keratinocytes (Kanemaru *et al.*, *Nat. Commun.* 2012 PMID: 22805570), we performed siRNA-mediated depletion of PLCδ1. PLCδ1 knockdown did not induce a loss of epithelial characteristics (Supplementary Fig. 4d in the revised manuscript).

4) The organization of the paper is not optimal – the effects of increasing PIP2 levels (Supplementary Figure 5) are only presented after the data about cholesterol levels (Figure 4), and the effects of changing PIP2 levels on E-cadherin localization are not shown until after the effect of cholesterol reduction on E-cadherin have been discussed. Figure 6, on the effects of PIP2 elevation on U2OS cells is not very interesting – if the authors want to argue that increased PIP2 will reduce tumorigenicity then they need to do in vivo assays to measure this. Effects on migration are not particularly novel and seem peripheral to the main arguments of the manuscript. I would suggest these data go into a Supplementary figure or be left out, and the effects of increased PIP2 on TGFβ-induced phenotypes be moved to the main figures.

We appreciate the reviewer's constructive suggestions. We have reorganized the manuscript, as suggested by the reviewer. Specifically, the data on the effects of increased PI(4,5)P₂ on TGFβ-induced phenotypes are now shown in a main figure (Fig. 4 in the revised manuscript). We also performed immunofluorescence experiments (Fig. 2 and Fig. 3 in the revised manuscript) of E-cadherin and other junctional and polarity proteins. The data on the effects of PI(4,5)P₂ elevation on migration, invasion, proliferation, and colony formation of U2OS cells are shown in the supplementary figure (Supplementary Fig. 7 in the revised manuscript). [Redacted]

5) Most importantly, the mechanistic aspects of the manuscript are rather superficial - the authors do not investigate the mechanism that ensures PIP₂ levels are higher in epithelia than in non-epithelial cells, except in response to TGFβ. They do not explore how TGFβ causes a decrease in PIP5K1A, or how reduced PIP₂ causes a decrease in PM cholesterol [Redacted]

According to the reviewer's suggestion, we have performed experiments and added data to explain the following mechanisms.

The mechanism that ensures PIP₂ levels are higher in epithelial cells than in non-epithelial cells

Since siRNA-mediated silencing of hPIP5K1A reduces PIP₂ levels (Fig. 3c and d in the revised manuscript), hPIP5K1A expression is essential for maintaining higher levels of PI(4,5)P₂ in the PM of epithelial cells compared to that in non-epithelial cells. However, mRNA expression level of PIP5K1A was not significantly higher in epithelial cells than in mesenchymal cells. Interestingly, hPIP5K1A-GFP was more remarkably localized to the PM in epithelial cells than in non-epithelial cells (Supplementary Fig. 3a and 3c in the revised manuscript). This observation suggests that the more prominent PM localization of hPIP5K1A likely contributes to higher PI(4,5)P₂ levels in epithelial cells.

Regarding the mechanisms by which TGFβ decreases PIP5K mRNA expression

The clarification of the detailed mechanisms by which TGFβ decreases mRNA expression of hPIP5K1A is beyond the scope of this study, but an important point for future research. Interestingly, TGFβ treatment not only decreased hPIP5K1A expression level, but also inhibited the localization of hPIP5K1A-GFP to the PM (Supplementary Fig. 3b and 3c in the revised manuscript), suggesting that impaired localization of hPIP5K1A to PM might also contribute to PI(4,5)P₂ reduction after TGFβ treatment.

Regarding the mechanisms by which PIP₂ regulates PM cholesterol

We examined the possibility that the total cholesterol amount is altered by PI(4,5)P₂ depletion by quantifying total cellular cholesterol and cholesterol ester levels. However, the amounts of cholesterol and cholesterol esters were not decreased (Fig.5c in the revised manuscript), strongly suggesting that PI(4,5)P₂ positively regulates PM transport of cholesterol or negatively regulates cholesterol extraction from the PM. A previous report showed that the loss of adherens junctions (AJ) decreases PM cholesterol (Shigetomi *et al.*, *J. Cell Biol.* 2018 PMID: 29720382). Given that PI(4,5)P₂ depletion

impairs the formation of the AJ (Fig. 2d and Fig. 3f in the revised manuscript), PI(4,5)P₂ affects PM cholesterol levels by regulating AJ formation. Since PM cholesterol depletion impaired AJ formation (Fig. 5h in the revised manuscript), PM accumulation of cholesterol and AJ formation would be interdependent events (p.22 l.7-13 in the revised manuscript).

[redacted]

6) Finally, the authors do not cite multiple prior papers describing a role for PIP₂ on epithelial polarity, including one showing that PI(4,5)P₂ controls apical size by tethering Par3 (Claret *Curr Biol* 2014), another that Crumbs regulates apical levels of PI(4,5)P₂ (Lattner *ELife* 2019), and one that vesicle exocytosis occurs at PI(4,5)P₂ rich domains (Martin *BBA* 2015).

We thank the reviewer for bringing these important papers to our attention. We referred to papers that the reviewer suggested (p.23 l.4-8, p.24 l.13-17 in the revised manuscript).

7) It seems likely, based on prior literature, that a key mechanism for the observed loss of epithelial characteristics is a loss of polarized vesicle transport (see for instance the Martin review), but this is not discussed at all.

The reviewer's comment is very insightful and it helped us come up with the concept that Par3 and exocytosis are critical for the maintenance or determination of epithelial characteristics by PI(4,5)P₂. Par3 regulates PM transport of E-cadherin by functioning as a receptor for exocysts in PM (Murkhtar Ahmed *et al.*, *Nat. Commun.* 2017 PMID: 28358000). We confirmed that Par3 knockdown impaired PM localization of E-cadherin and other junctional proteins in HaCaT cells (Fig. 6a-c in the revised manuscript), similar to how PI(4,5)P₂ depletion did. We also found that PI(4,5)P₂ is required for maintenance of PM Par3 levels in HaCaT cells (Fig. 2d, e and Fig. 3f, g in the revised manuscript). In addition, we showed that elevation of PI(4,5)P₂ levels in osteosarcoma cells increased the Par3 level in the PM (Fig. 7g, h in the revised manuscript). Since Par3 binds to PI(4,5)P₂ (Horikoshi *et al.*, *Cell Struct. Funct.* 2011 PMID: 21467691) and PM Par3 functions as the receptor for exocyst (Murkhtar Ahmed *et al.*, *Nat. Commun.* 2017), PI(4,5)P₂ likely recruits Par3 to the PI(4,5)P₂-rich PM, and Par3 functions as the receptor for exocyst in PI(4,5)P₂-rich PM. Interestingly, expression of the exocyst-docking region of Par3 fused with PM-targeting peptide (Lyn-hPar3 (710–1089)-V5) provided osteosarcoma cells with partial epithelial characteristics (Fig. 7j-l in the revised manuscript) in a manner similar to the elevation of PI(4,5)P₂ levels by Lyn-mPIP5Kwt-GFP expression. Collectively, these results suggest that PI(4,5)P₂ likely maintains epithelial characteristics by positively regulating PM localization of Par3 and trafficking of E-cadherin and other junctional proteins to the PM (Supplementary Fig. 8. in the revised manuscript).

Reviewers' Comments:

Reviewer #1:

Remarks to the Author:

The authors had addressed my concerns and I have no further questions.

Reviewer #2:

Remarks to the Author:

The authors have adequately addressed my comments. The paper is now suitable for publication in Nature Communications.

Reviewer #3:

Remarks to the Author:

Overall I feel that the authors have done an exceptionally thorough revision, with extensive new data, and I consider the manuscript now suitable for publication with out any further major revisions. However, I do have a minor problem with the way the data are expressed - apparently throughout the study the authors performed only 2 biological replicates. They show all of the technical replicate data (measurements on single cells or fields of view) in the box plots, which is great, but they do not distinguish which points are from which biological replicate so there is no way of knowing what the true variance is between these replicates. I feel that this is not appropriate and that they need to distinguish the biological replicates in each of the plots. They should also indicate the value of N for biological replicates in each figure legend.

I note also that there is no quantification of the immunoblots, and apparently these were only performed once (!!) as there is no indication of replicates in the Methods section.

Responses to Reviewer 3

Overall I feel that the authors have done an exceptionally thorough revision, with extensive new data, and I consider the manuscript now suitable for publication without any further major revisions. However, I do have a minor problem with the way the data are expressed - apparently throughout the study the authors performed only 2 biological replicates. They show all of the technical replicate data (measurements on single cells or fields of view) in the box plots, which is great, but they do not distinguish which points are from which biological replicate so there is no way of knowing what the true variance is between these replicates. I feel that this is not appropriate and that they need to distinguish the biological replicates in each of the plots. They should also indicate the value of N for biological replicates in each figure legend. I note also that there is no quantification of the immunoblots, and apparently these were only performed once (!!) as there is no indication of replicates in the Methods section.

As the reviewer indicated, we performed two independent experiments for analysis of cell area and subcellular localization of target proteins. As suggested by the reviewer, we have differentiated the data from the first experiment and that from the second experiment in each of the plots. We also have added the following description regarding biological replicates according to the editorial guidance in the author checklist: "X cells were examined over 2 independent experiments." (X is the number of cells).

We apologized for the insufficient information on the immunoblots. We performed two independent experiments and obtained similar results from them. Therefore, we have added the description "Immunoblot data shown are representative of two independent experiments with similar results" to the figure legends. Since target proteins were hardly detected in some samples including inhibitor- or siRNA-treated samples and it is difficult to quantify protein levels accurately, we did not perform quantification.